

# Assessing the formation and evolution mechanisms of severe haze pollution in Beijing−Tianjin−Hebei region by using process analysis

Lei Chen[1,2], Jia Zhu[1], Hong Liao[1], Yi Gao[3], Yulu Qiu[4,5], Meigen Zhang[3,6,7] and Nan Li[1]

[1]Jiangsu Key Laboratory of Atmospheric Environment Monitoring and Pollution Control, Jiangsu Collaborative Innovation Center of Atmospheric Environment and Equipment Technology, School of Environmental Science and Engineering, Nanjing University of Information Science & Technology, Nanjing, 210044, China

[2]Key Laboratory of Meteorological Disaster, Ministry of Education (KLME), Joint International Research Laboratory of Climate and Environment Change (ILCEC), Collaborative Innovation Center on Forecast and Evaluation of Meteorological Disasters (CIC-FEMD), Nanjing University of Information Science & Technology, Nanjing, 210044, China

[3]State Key Laboratory of Atmospheric Boundary Layer Physics and Atmospheric Chemistry, Institute of Atmospheric Physics, Chinese Academy of Sciences, Beijing, 100029, China

[4]Institute of Urban Meteorology, China Meteorological Administration, Beijing 100089, China

[5]Beijing Shangdianzi Regional Atmosphere Watch Station, Beijing, China

[6]University of Chinese Academy of Sciences, Beijing, 100049, China

[7]Center for Excellence in Regional Atmospheric Environment, Institute of Urban Environment, Chinese Academy of Sciences, Xiamen, China

*Correspondence to*: H. Liao (hongliao@nuist.edu.cn)

**Abstract.** Fine–particle pollution associated with haze threatens human health, especially in the North China Plain, where extremely high $PM_{2.5}$ concentrations were frequently observed during winter. In this study, the WRF–Chem model coupled with an improved integrated process analysis scheme was used to investigate the formation and evolution mechanisms of a haze event happened over Beijing–Tianjin–Hebei (BTH) in December 2015, including examining the contributions of local emission and outside transport to the absolute $PM_{2.5}$ concentration in BTH, and the contributions of each detailed physical or chemical process to the variations in the $PM_{2.5}$ concentration. The influence mechanisms of aerosol radiative forcing (including aerosol direct and indirect effects) were also examined by using the process analysis. During the aerosol accumulation stage (December 20–22, Stage_1), the average near–surface $PM_{2.5}$ concentration in BTH was 250.0 μg m$^{-3}$, which was contributed by local emission of 42.3% and outside transport of 36.6%. During the aerosol dispersion stage



(December 23–27, Stage_2), the average concentration of $PM_{2.5}$ was 107.9 μg m$^{-3}$. The contribution of local emission increased to 50.9%, while the contribution of outside transport decreased to 24.3%. The 24–h change (23:00LST minus 00:00LST) in the near–surface $PM_{2.5}$ concentration was +50.4 μg m$^{-3}$ during Stage_1 and −41.5 μg m$^{-3}$ during Stage_2. Contributions of aerosol chemistry process and vertical mixing process to the 24–h change were +43.8 (+17.9) μg m$^{-3}$ and −161.6 (−221.6) μg m$^{-3}$ for Stage_1 (Stage_2), respectively. Small differences in contributions from other processes were found between Stage_1 and Stage_2, such as advection process, cloud chemistry process, and so on. Therefore, the $PM_{2.5}$ increase over BTH during haze formation stage (Stage_1) was mainly attributed to strong production by aerosol chemistry process and weak removal by vertical mixing process. When aerosol radiative feedback was considered, the 24–h $PM_{2.5}$ increase was enhanced by 9.6 μg m$^{-3}$ during Stage_1, which could be mainly attributed to the contributions of vertical mixing process (+39.8 μg m$^{-3}$), advection process (−38.6 μg m$^{-3}$) and aerosol chemistry process (+5.1 μg m$^{-3}$). The restrained vertical mixing could be the primary reason for the enhancement in near–surface $PM_{2.5}$ increase when aerosol radiative forcing was considered.

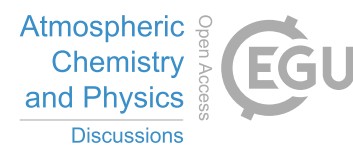
# 1 Introduction

Anthropogenic activities associated with rapidly developed industrialization and urbanization have been leading to a sustained increase in the amounts of atmospheric pollutants, especially in the fast–developing countries (IPCC, 2013). As one of the largest emission sources of aerosols and their precursors, China has been suffering from serious air pollution for

years (Lei et al., 2011; Li et al., 2011), with severe haze events frequently occurring in winter, especially over large urban agglomerations, such as the North China Plain (NCP) (Han et al., 2014; Gao et al., 2015), the Yangtze River Delta area (YRD) (Ding et al., 2016; Wang et al., 2016a), the Pearl River Delta area (PRD) (Fan et al., 2015; Liu et al., 2018b), and the Sichuan Basin (SCB) (Zhao et al., 2018; Zhang et al., 2019). During severe haze events, the observed maximum hourly surface–layer $PM_{2.5}$ (fine particulate matter with aerodynamic diameter of 2.5 μm or less) concentration exceeded 1000 μg

$m^{-3}$ (Wang et al., 2013b; Sun et al., 2016; Li et al., 2017a), which could significantly influence visibility (Li et al., 2014), radiation budget (Steiner et al., 2013), atmospheric circulation (Jiang et al., 2017), cloud properties (Unger et al., 2009), and even human health (Guo et al., 2017).

Extensive studies have been carried out in recent years to analyze the formation mechanisms of haze episodes in China. Wang et al. (2013a) used a synergy of ground–based observations, satellite, and lidar measurements to study a long–lasting

and severe haze episode occurred in eastern China in January 2013, and concluded that stagnant meteorological conditions, which could be generally characterized by weak wind speed, high relative humidity, intense inversion, and low mixing layer height, were tightly associated with severe haze episodes. Based on National Center for Environmental Prediction (NCEP) reanalysis data, Shu et al. (2017) identified five typical synoptic patterns, and pointed out that each synoptic pattern exerted different impacts on particle pollution over YRD. By analyzing the simulation results from a large ensemble climate model

(MIROC5), Li et al. (2018) investigated the contributions of anthropogenic influence to severe haze events happened over eastern China in January 2013 and December 2015, and found that anthropogenic forcing (i.e., increased emissions of greenhouse gases) could modify atmospheric circulation pattern, and these human–induced circulation changes were conducive to the occurrence of severe haze events. Zhang et al. (2015a) used a global 3–D chemical transport model (GEOS–Chem) to quantify the local source contributions to wintertime surface–layer $PM_{2.5}$ concentrations over North China

from 2013 to 2015, and reported that emissions from residential and industrial sources and transportation contributed most to



the high concentrations of atmospheric aerosols in Beijing. Many studies reported that regional transport of aerosols also played an important role in haze episodes (Wang et al., 2013b; Jiang et al., 2015; Zheng et al., 2015). Wang et al. (2013b) reported that the cross–city clusters transport outside BTH (Beijing, Tianjin, and Hebei) and transport among cities inside BTH contributed 20%–35% and 26%–35% of $PM_{2.5}$ concentrations over BTH, respectively. Secondary aerosol formation

and their hygroscopic growth were also confirmed to be a large contribution to severe haze episodes (Huang et al., 2014b; Han et al., 2015; Chen et al., 2019). The conversion of $SO_2$ to $SO_4^{2-}$ was strongly associated with high relative humidity, and $NO_3^-$ was found to be produced mainly by photochemical and heterogeneous reactions (Chen et al., 2016; Zhang et al., 2018a).

It is well known that aerosols can scatter and absorb solar radiation to alter the radiative balance of the atmosphere and

surface (direct radiative effect), and can serve as cloud condensation nuclei or ice nuclei to affect cloud properties (indirect radiative effect) (Twomey, 1974). These impacts are coupled with atmospheric dynamics to produce a chain of interactions with a large range of meteorological variables that influence both weather and climate (Ramanathan et al., 2001; Huang et al., 2006; Wu et al., 2009; Li et al., 2017d), which will further induce feedbacks on aerosol production, accumulation, and even severe haze pollutions (Petaja et al., 2016; Li et al., 2017b; Li et al., 2017e; Zhao et al., 2017; Gao et al., 2018). Based on

multi–year measurements (from 2010 to 2016), Huang et al. (2018) found that aerosol radiative effects led to a significant heating in the upper planetary boundary layer (PBL) and a substantial dimming at the surface over North China. This is because high concentrations of light–absorbing aerosols were observed, and the aerosol–meteorology interactions depressed the development of PBL, and therefore aggravated the haze pollution (Su et al., 2018). By using the WRF–Chem model, Gao et al. (2015) analyzed the feedbacks between aerosols and meteorological fields over NCP in January 2013, and found that

aerosols caused a significant negative (positive) radiative forcing at the surface (in the atmosphere), resulting in a weaker surface–layer wind speed and lower PBL height (PBLH). The average surface–layer $PM_{2.5}$ concentration was increased by 10–50 µg m$^{-3}$ as a result of the more stable atmosphere. By analyzing the observations from a comprehensive field experiment and simulation results from WRF–Chem model, Liu et al. (2018a) concluded that the decreased PBLH associated with increased aerosol concentrations could enhance surface–layer relative humidity by weakening the vertical transport of

water vapor, and the increased relative humidity at the surface accelerated the formation of secondary particulate matters



(SPM) through heterogeneous reactions, leading to the increase of the $PM_{2.5}$ concentration by 63 μg m$^{-3}$ averaged over the NCP during 15–21 December, 2016.

All these studies discussed above revealed that the formation of haze episodes was caused by the interactions between local emissions, regional transport, meteorological conditions, and chemical production. Nevertheless, only the net combined

effects on the concentrations of pollutants were provided, without the capabilities of understanding and isolating the atmospheric physical and chemical processes involved. The quantitative assessment of the contributions from each detailed physical/chemical process (e.g., vertical mixing process, advection process, emission source process, aerosol chemistry process, cloud chemistry process, and so on) is necessary for fully understanding of the formation and evolution mechanisms of haze episodes (Goncalves et al., 2009; Xing et al., 2017; Kang et al., 2019). What's more, although many previous studies

have identified the positive feedback effects of aerosol radiative forcing on particulate accumulation, the detailed influence mechanisms of the forcing–response relationship at each process chain remain largely elusive (i.e., the prominent physical or chemical processes responsible for the aerosol radiative impacts on haze episodes). Since 2009, substantial efforts have been taken to improve air quality in China, including emission reduction and energy transition. However, haze events continued to occur frequently all over the country. For example, a severe, long–lasting, and wide–ranging haze episode was observed in

December 2015 over the central and eastern China, with the regional average $PM_{2.5}$ concentration exceeding 150 μg m$^{-3}$. For BTH, a red alert for haze (the most serious level) was issued for the period from 20 to 22 December 2015, with the maximum hourly $PM_{2.5}$ concentration exceeding 1000 μg m$^{-3}$. The formation and evolution mechanisms, and the aerosol radiative feedbacks of this severe haze episode have not been fully estimated yet.

In this study, we develop an improved online integrated process rate (IPR) analysis scheme (i.e., process analysis) in the

fully coupled online Weather Research and Forecasting–Chemistry (WRF–Chem) model, to investigate the formation and evolution mechanisms of the severe haze episode happened over NCP from 20 to 29 December 2015. Sensitivity experiments are conducted to examine the contributions of local emission and outside transport to the absolute $PM_{2.5}$ concentrations during the haze episode, while the IPR analysis is used to quantify the contributions of each detailed physical/chemical process to the variations in the $PM_{2.5}$ concentrations. The effects of aerosol radiative forcing, including

direct and indirect effects, on meteorological parameters and $PM_{2.5}$ levels during the haze episode are also quantified, with a

special focus on the detailed influence mechanism. We hope that the results concluded in this study may provide better understanding of the formation mechanisms for severe haze events, and help policy makers take targeted measures to improve air quality over North China.

This manuscript is arranged as follows. Model configuration, integrated process rate (IPR) analysis (i.e., process analysis), numerical experiments, and observations are presented in Section 2. Model evaluation is conducted in Section 3. The formation and evolution mechanisms of the haze episode are investigated in Section 4. Section 5 provides the impacts of aerosol radiative forcing. Summaries and discussions are presented in Section 6.

## 2 Methods

### 2.1 Model configuration

A fully coupled online Weather Research and Forecasting–Chemistry model (WRF–Chem v3.7) is used to simulate meteorological fields and concentrations of gases and aerosols simultaneously (Skamarock et al., 2008; Grell et al., 2005). The WRF–Chem model is designed with two domains using 219 (west–east) ×159 (south–north) and 150 (west–east) ×111 (south–north) grid points at the horizontal resolutions of 27 and 9 km, respectively (Fig. 1). The outer domain covers nearly the whole East Asia, and the inner domain is located in the NCP. In order to minimize the impacts from IBCs (lateral boundary conditions), we only analyze the simulation results from the inner region of the second domain (i.e., BTH), following Chen et al. (2018) and Wu et al. (2012). The vertical dimension is resolved by 29 full sigma levels, with 16 layers located in the lowest 2 km for finer resolution in the planetary boundary layer, and the height of the first layer averaged in BTH is about 30 m.

Meteorological initial and lateral boundary conditions used in the WRF–Chem model are taken from the NCEP (National Center for Environmental Prediction) Final Operational Global Analysis data with the spatial resolution of $1° \times 1°$. Four–dimensional data assimilation (FDDA) with the nudging coefficient of $3.0 \times 10^{-4}$ for wind (in and above PBL), temperature (above PBL) and water vapor mixing ratio (above PBL) is adopted to improve the accuracy of simulation results (Lo et al., 2008; Otte, 2008; Wang et al., 2016b). The forecasts from the global chemical transport model MOZART–4 are



processed to provide the chemical initial and boundary conditions for the WRF–Chem model (Emmons et al., 2010).

Anthropogenic emission data are obtained from the MIX Asian emission inventory (http://www.meicmodel.org/dataset-mix.html), with a horizontal resolution of 0.25 degree (Li et al., 2017c). It is developed to support the MICS–Asia III (Model Inter–Comparison Study for Asia Phase III) and the TF HTAP (Task Force on Hemispheric Transport of Air Pollution) projects. This inventory includes $SO_2$ (sulfur dioxide), $NO_x$ (nitrogen oxides), CO (carbon monoxide), $CO_2$ (carbon dioxide), NMVOC (non–methane volatile organic compounds), $NH_3$ (ammonia), BC (black carbon), OC (organic carbon), $PM_{2.5}$ and $PM_{10}$. All these species are from several sectors, such as agriculture, industry, power, transportation and residential, and the emission rate of each species for each hour is based on Gao et al. (2015). The biogenic emissions are calculated online using the MEGANv2.04 (Model of Emission of Gases and Aerosol from Nature v2.04) model (Guenther, 2006). Biomass–burning emissions are obtained from the GFEDv3 (Global Fire Emissions Database v3) (Randerson et al., 2005). Dust emissions and sea salt emission are calculated online by using algorithms proposed by Shao (2004) and Gong et al. (1997), respectively.

The Carbon–Bond Mechanism version Z (CBMZ) (Zaveri and Peters, 1999) is selected to simulate the gas phase chemistry, and the 8–bin sectional aerosol module, MOSAIC (Model for Simulating Aerosol Interactions and Chemistry) (Zaveri et al., 2008), with some aqueous chemistry, is used to simulate aerosol evolution. All major aerosol species are considered in the MOSAIC scheme, including $SO_4^{2-}$, $NO_3^-$, $NH_4^+$, Cl, Na, BC, primary organic mass, liquid water, and other inorganic mass (Zaveri et al., 2008). The aerosol size distribution is divided into discrete size bins defined by their lower and upper dry particle diameters (Zhao et al., 2010). In the current CBMZ/MOSAIC scheme, the formation of SOA (secondary organic aerosol) is not included (Zhang et al., 2012; Gao et al., 2016). Aerosol optical properties, including extinction efficiency, single scatter albedo, and asymmetry factor are computed by Mie theory, based on aerosol composition, mixing state, and size distribution (Barnard et al., 2010). The impacts of aerosols on photolysis rates are calculated using the Fast–J photolysis scheme (Wild et al., 2010). Aerosol radiation is simulated by RRTMG (Rapid Radiative Transfer Model for GCMs) for both shortwave (SW) and longwave (LW) radiation (Zhao et al., 2011). More information about the parameterizations used in this study can be found in Table 1.



## 2.2 Integrated process rate (IPR) analysis

Most air quality models are configured to output only the pollutant concentrations that reflect the combined effects of all physical and chemical processes. Quantitative information of the impacts of individual process is usually unavailable. Process analysis techniques (i.e., integrated process rate (IPR) analysis) can be used in grid–based Eulerian models (e.g.,

WRF-Chem) to obtain contributions of each physical/chemical process to variations in pollutant concentrations. Eulerian models utilize the numerical technique of operator splitting to solve continuity equations for each species into several simple ordinary differential equations or partial differential equations that only contain the influence of one or two processes (Gipson, 1999).

The IPR analysis method has been fully implemented in Community Multi–scale Air Quality (CMAQ) model, and has

been widely applied to study regional photochemical ozone ($O_3$) pollution (Goncalves et al., 2009; Khiem et al., 2010; Xing et al., 2017; Tang et al., 2017). Several WRF–Chem model studies used the IPR analysis to investigate the impacts of physical/chemical process on variations in $O_3$ concentrations. Gao et al. (2018) investigated the impacts of BC–PBL interactions on $O_3$ concentrations by analyzing the contributions from photochemistry, vertical mixing, and advection processes. Jiang et al. (2012) calculated the contributions of photochemical reactions and physical processes to $O_3$ formation

by using a simplified IPR analysis scheme.

Applying the IPR analysis to diagnose the contributions of each physical or chemical process to variations in aerosol concentrations in WRF–Chem model is more complex technically, and therefore few studies conducted the IPR analysis for aerosols. In this study, we developed an improved IPR analysis scheme in the WRF–Chem model to isolate the processes impacting variations in aerosol concentrations into nine different processes, namely advection (TRAN), emission source

(EMIS), dry deposition (DYRD), turbulent diffusion (DIFF), sub–grid convection (SGCV), gas–phase chemistry (GASC), cloud chemistry (CLDC), aerosol chemistry (AERC), and wet scavenging (WETP). DRYD is based on resistance models for trace gases (Wesely, 1989) and aerosol particles (Ackermann et al., 1998). SGCV includes the scavenging and aqueous chemistry within the wet convective updrafts. WETP contains in–cloud rainout and below–cloud washout during grid–scale precipitation. The contribution of individual process can be calculated as the difference of aerosol concentrations before and

after the corresponding operator.



Base on the principle of mass balance, IPR can be verified by comparing the variations in aerosol concentrations (the concentration at the current time minus the concentration at the previous time) with the sum of the contributions from the nine processes during each time step. As shown in Fig. S1, the net contributions of all processes match the variations in aerosol concentrations pretty well.

## 2.3 Numerical experiments

Table 2 summarizes the experimental design. To investigate the contributions of outside transport and local emission to the absolute $PM_{2.5}$ concentrations in BTH, four simulations with different anthropogenic emission categories are conducted: (1) CTL: The control simulation with all anthropogenic emissions considered; (2) NoAnth: No anthropogenic emission is considered in the whole domain; (3) NoBTH_Anth: Same as CTL, but anthropogenic emissions in BTH are excluded; (4) OnlyBTH_Anth: Contrary to the NoBTH_Anth case, anthropogenic emissions are only considered in BTH. All the physical and chemical schemes used in these cases are identical. The contributions of regional transport and local emission to the absolute $PM_{2.5}$ concentration in BTH can be identified by comparing the simulation results of NoBTH_Anth and NoAnth (i.e., NoBTH_Anth minus NoAnth) and OnlyBTH_Anth and NoAnth (i.e., OnlyBTH_Anth minus NoAnth), respectively.

The IPR analysis method is applied to two stages in the CTL case to quantify the contribution of each detailed physical and/or chemical process to the variations in the $PM_{2.5}$ concentration. Comparing the contribution of each process between the two stages can quantitatively explain the reason for $PM_{2.5}$ increase during the stage of haze accumulation and $PM_{2.5}$ decrease during the stage of haze dispersal.

To quantify the aerosol radiative effects (ARE) on the haze episode, another sensitivity experiment (referred to as NoARE case) is designed by closing the feedbacks between aerosols and meteorological variables, including eliminating the aerosol direct effect (ADE) and aerosol indirect effect (AIE) in the model. The ADE is turned off by removing the mass of aerosol species from the calculation of aerosol optical properties as did in Qiu et al. (2017). The AIE is turned off by using the constant cloud droplet number concentration (CDNC), which is calculated from the CTL case during the whole simulation period, following Gao et al. (2015). The differences between CTL and NoARE (i.e., CTL minus NoARE) represent the impacts of aerosol radiative forcing.



The IPR analysis method is then applied to CTL and NoARE cases, respectively, to investigate the detailed influence mechanisms (i.e., the prominent physical or chemical process responsible for the aerosol radiative impacts on the haze episode).

All the five simulations are conducted for the period from 17 to 29 December 2015, and the initial three days are discarded as the model spin–up to minimize the impacts of initial conditions. Simulation results from the CTL case during 20 to 29 December 2015 are used to evaluate the model performance.

**2.4 Observational data**

Simulated meteorological parameters in CTL case, including 2 m temperature ($T_2$), 2 m relative humidity ($RH_2$), 10 m wind speed ($WS_{10}$) and 10 m wind direction ($WD_{10}$), are compared with hourly observations at 12 stations, which are collected from NOAA's National Climatic Data Center (https://gis.ncdc.noaa.gov/maps/ncei/cdo/hourly). The meteorological observation sites are marked in blue dots in Fig. 1(b). PBL height (PBLH) in 3–hour intervals provided by NOAA READY archived meteorological GDAS data (https://ready.arl.noaa.gov/READYamet.php) in Beijing (39.93 °N, 116.28 °E) (marked in purple dot in Fig. 1(b)) is also used to evaluate the model performance. Hourly shortwave downward radiation flux (SWDOWN) at the Xianghe station (39.75 °N, 116.96 °E), marked in light green dot in Fig. 1(b), is taken from WRMC–BSRN (World Radiation Monitoring Center–Baseline Surface Radiation Network, http://bsrn.awi.de) for the energy budget evaluation. The hourly observed surface–layer $PM_{2.5}$ concentrations at the 59 stations (marked in red dots in Fig. 1(b)) are obtained from the CNEMC (China National Environmental Monitoring Center, http://www.cnemc.cn/). The daily measurements of mass concentrations of $SO_4^{2-}$, $NO_3^-$, $NH_4^+$, BC and OC are also collected at the two sites in Beijing (39.97 °N, 116.37 °E) and Shijiazhuang (38.03 °N, 114.53 °E) (marked in dark green triangles in Fig. 1(b)).

**3. Model evaluation**

Accurate representations of observed meteorological fields and pollutant concentrations provide foundations for haze analysis with the WRF–Chem model. Detailed comparisons between observed and simulated meteorological parameters ($T_2$, $RH_2$, $WS_{10}$, $WD_{10}$, PBLH, and SWDOWN) and pollutant concentrations ($PM_{2.5}$, BC, OC, $SO_4^{2-}$, $NO_3^-$, and $NH_4^+$) are

presented in this section.

## 3.1 Meteorological parameters

Figure 2 shows the time series of observed and simulated hourly meteorological variables averaged over the 12 stations during 20–29 December 2015. Corresponding statistical metrics, including mean value, mean bias (MB), gross error (GE), normalized mean bias (NMB), root mean square error (RMSE), mean fractional bias (MFB), mean fractional error (MFE), index of agreement (IOA), and correlation coefficient (R) are presented in Table 3. As shown in Fig. 2, simulated $T_2$, $RH_2$, $WS_{10}$ and $WD_{10}$ agree well with the observational data. For temperature, WRF–Chem model can perfectly depict its diurnal and daily variations with both R and IOA of 0.9, but slightly overestimates the low values at night, with the NMB of 0.4%. Observed relative humidity can be reasonably reproduced by the model with R and IOA of 0.8 and 0.7, respectively. But a persistent underestimation is found with the NMB of −14.3%. Different surface layer and boundary layer options may have influence on the simulated near–surface moisture fluxes, and the settings of these schemes can partially explain the biases of $RH_2$ between observations and simulations (Qian et al., 2016). This negative bias of $RH_2$ was also reported in other studies (Zhang et al., 2009; Gao et al., 2015). WRF–Chem model can capture the observed low values of wind speed during 20–22 December and high values of wind speed during 25–27 December. The positive NMB of 29.1% may probably result from unresolved topographical features in surface drag parameterization and the coarse resolution used in the nested domain (Yahya et al., 2015; Zheng et al., 2015a). For wind direction, the calculated NMB is −1.3% and the R is 0.6, indicating that the WRF–Chem model can generally reproduce the varied wind direction during the simulation period.

Simulated hourly PBLH and SWDOWN are also compared with observations in Fig. 3. It is noted that PBLH provided by GDAS of NOAA are in 3–hour intervals. The simulations in CTL case agree well with the observations, including capturing the daily maximum in the daytime and the low values at night. The correlation coefficients are 0.7 and 0.9 for PBLH and SWDOWN, respectively.

## 3.2 $PM_{2.5}$ and its components

Observed hourly surface–layer $PM_{2.5}$ concentrations from 20 to 29 December 2015 in the nine cities (Shengyang,





Beijing, Xingtai, Hengshui, Baoding, Langfang, Yangquan, Anyang, and Jinan) are compared with the model results from CTL case (Fig. 4). The statistical metrics are shown in Table 3. Generally, WRF–Chem model can reasonably reproduce the evolutional characteristics of the observed $PM_{2.5}$ concentrations in the nine cities (Rs=0.58–0.88). Both the observed and simulated $PM_{2.5}$ concentrations exhibit a growth trend during 20–22 and 28–29 December, and a decreasing tendency during

23–27 December. However, an obvious underestimation is found in Beijing from 25 to 26 December when a maximum hourly concentration of 600 μg m$^{-3}$ was observed. The negative bias is also simulated by previous studies (Chen et al., 2018; Zhang et al., 2018b), and the possible reasons for the underestimation are (1) the bias in simulated meteorological conditions (e.g., underestimated $RH_2$ and overestimated $WS_{10}$); (2) the missing mechanisms of some gas–aerosol phase partitioning and heterogeneous reactions which may produce secondary inorganic aerosol (Huang et al., 2014a; Wang et al., 2014); (3) the

lack of SOA simulation in MOSAIC mechanism (Gao et al., 2016). Generally, the performance statistics of $PM_{2.5}$ in almost all cities meet the model performance goal (MFB within ±30% and MFE ≤ 50%) proposed by Boylan and Russel (2006).

Figure 5 compares the simulated and observed surface–layer concentrations of BC, OC, $SO_4^{2-}$, $NO_3^-$, and $NH_4^+$ in Beijing and Shijiazhuang averaged during 20–29 December 2015. WRF–Chem model underestimates the BC, OC, $NH_4^+$ and $SO_4^{2-}$ concentrations in Beijing (Shijiazhuang) by 3.6% (33.1%), 34.9% (38.6%), 24.1% (37.5%), and 32.5% (44.6%),

respectively, and overestimates the concentrations of $NO_3^-$ by 22.2% (51.8%). Due to the low reactivity of BC in the atmosphere, the uncertainty in BC emission may cause this underestimation (Li et al., 2017c). For OC, the underestimation may result from the lack of SOA in the MOSAIC aerosol module (Qiu et al., 2017). Missing some mechanisms of $SO_2$ gas–phase and aqueous–phase oxidation, as well as heterogeneous chemistry may explain the underestimation of $SO_4^{2-}$. It is noted that the biases of the aerosol components were also reported by other WRF–Chem studies (Zhang et al., 2015a; Qiu et

al., 2017).

## 4. Formation and evolution mechanisms of the haze episode

In this section, we first reproduce the evolution of this severe haze episode, and then investigate the formation and evolution mechanisms, including examining contributions of local emission and outside transport to the absolute $PM_{2.5}$ concentrations, and the contributions of each detailed physical/chemical process to the variations in the $PM_{2.5}$ concentrations.



### 4.1 Spatial-temporal evolutions of surface-layer PM$_{2.5}$ concentrations

Figures 6(a-j) show the spatial distributions of simulated daily mean surface-layer PM$_{2.5}$ concentrations from 20 to 29 December 2015. On December 20, the BTH region was under a uniform pressure field (Fig. S2(a)). The regional average wind speed was less than 3 m s$^{-1}$, and the boundary layer became stable, which constrained aerosols within a low mixing layer. Meanwhile, a low-pressure center situated to the north of BTH, where air pollutants from south, southwest, and southeast converged. Consequently, the daily mean PM$_{2.5}$ concentration averaged over BTH was over 200 μg m$^{-3}$. On December 21, a weak low-pressure center was formed near the Bohai Bay and a weak high-pressure center moved to Shandong Peninsula (Fig. S2(b)). The synoptic conditions brought more air masses from south to north, and worsened air quality in BTH. On December 22, a weak high pressure system moved within Inner Mongolia (Fig. S2(c)), which could bring cold air to the BTH region. Meanwhile, the polluted air could also be transported back to the BTH, leading to a continuous increase in the PM$_{2.5}$ concentrations, with the maximum daily mean concentration exceeding 600 μg m$^{-3}$ in BTH (Fig. 6(c)). Due to the enhanced anticyclone originated from Siberian (Fig. S2(d)), the accumulation of aerosol particles in BTH was terminated with the incursion of a strong cold front from 23 to 27 December. But frequent transitions between high and low pressure systems over BTH accompanying with the shifting wind directions resulted in a quick PM$_{2.5}$ variation, especially on December 24 and 25, when a low-pressure system developed northeast of BTH (Fig. S2(e)). The air mass in BTH was influenced by the pollutants from south, resulting in a temporary increase in the concentration of PM$_{2.5}$ on December 25. On December 28 and 29, another haze episode occurred.

According to the trends in simulated PM$_{2.5}$ concentrations averaged over the BTH region (Figs. 6(k-l)), we divide the whole simulation period into three stages: (1) aerosol accumulation stage (December 20-22, Stage_1); (2) aerosol dispersion stage (December 23-27, Stage_2); (3) formation stage for another haze event (December 28-29, Stage_3). In this manuscript, we mainly focus on the first two stages to reveal important factors that cause the accumulation and dispersion of particulate matters.

In Stage_1, the daily mean PM$_{2.5}$ concentrations averaged over BTH increased from 209.0 μg m$^{-3}$ to 289.8 μg m$^{-3}$, and the average PM$_{2.5}$ concentration was 250.0 μg m$^{-3}$ (Fig. 7(a)), far beyond the threshold value of "heavily polluted" (PM$_{2.5}$ 24-h average concentration > 150 μg m$^{-3}$). The WS$_{10}$ was 2.3 m s$^{-1}$ (Fig. 7(b)), less than 3.2 m s$^{-1}$ (one of the indicators used

to define air stagnation by NOAA, https://www.ncdc.noaa.gov/societal-impacts/air-stagnation/overview), indicating that the near surface circulation was insufficient to disperse accumulated air pollutants. The decreased PBLH (from 148.6 m to 109.9 m) could compress air pollutants into a shallow layer, resulting in an elevated pollution level. During Stage_2, the $PM_{2.5}$ concentrations decreased gradually with the increased wind speed and PBLH. The $PM_{2.5}$ concentration averaged during Stage_2 was 107.9 μg m$^{-3}$, still exceeding the Grade II standard (75 μg m$^{-3}$) defined by the National Ambient Air Quality Standards of China.

## 4.2 Contributions of local emission and regional transport to absolute $PM_{2.5}$ concentrations

Previous studies have reported that anthropogenic emission was the internal cause of haze events in China (Jiang et al., 2013; Sun et al., 2014; Gu and Liao, 2016; Yang et al., 2016b). Emission control measures have been taken to ensure good air quality for major events (e.g., APEC) or to mitigate the severity of coming pollution episodes (Zhou et al., 2018). Other studies, such as Sun et al. (2017) and Wang et al. (2017b), pointed out that outside transport contributed more than 50% of the particulate concentrations in BTH during haze events. This section discusses contributions of local emission and regional transport to the $PM_{2.5}$ concentrations in BTH, aiming to reveal the relative importance during this haze episode.

As shown in Fig. 7(a), the $PM_{2.5}$ concentrations during Stage_1 were contributed by the combined effects of local emission and regional transport. The contributions of local emission and regional transport were comparable (42.3% and 36.6%, respectively). In Stage_2, the contribution of outside transport decreased from 30.0% to 16.3%. The relative high $PM_{2.5}$ concentration was principally caused by local emission. On average, the contributions of local emission and regional transport during Stage_2 were 50.9% and 24.3%, respectively. The impact of outside transport could be qualitatively expressed by specific humidity, which was treated as an indicator for the origin of air masses (Jia et al., 2008). Air masses from the south were usually warmer and wetter than those from the north, so the specific humidity averaged over the BTH was higher in Stage_1 than that in Stage_2 (Fig. 7(b)). The evolution of $PM_{2.5}$ nicely followed the trend of specific humidity with a high correlation coefficient of 0.91.



### 4.3 Contributions of each physical/chemical process to variations in PM$_{2.5}$ concentrations

Figures 8(a1-a2) show the diurnal variations of PM$_{2.5}$ concentrations averaged over the BTH region during Stage_1 and Stage_2, respectively. The PM$_{2.5}$ concentration increased by 50.4 μg m$^{-3}$ (from 237.0 μg m$^{-3}$ at 00:00LST to 287.4 μg m$^{-3}$ at 23:00LST) during the period of particulate accumulation (Stage_1), but it decreased by 41.5 μg m$^{-3}$ during the period of particulate elimination (Stage_2).

The hourly PM$_{2.5}$ changes induced by each and all physical/chemical processes during Stage_1 and Stage_2 by using the IPR analysis method are shown in Figs. 8(b1-b2). During both stages, the dominant sources of surface-layer PM$_{2.5}$ were EMIS and AERC, while the main sinks were TRAN, DIFF, and DRYD. The maximum positive contribution of EMIS could be found during the rush hours (07:00-08:00LST and 16:00-19:00LST) (Fig. S3). The maximum negative contributions of TRAN and DIFF appeared at late night (00:00-05:00LST) and at noon (11:00-14:00LST), respectively.

To explain the reason for 24-h PM$_{2.5}$ increase during Stage_1 and 24-h PM$_{2.5}$ decrease during Stage_2 (Figs. 8(a1-a2)), we quantify the contributions of each physical/chemical process to 24-h PM$_{2.5}$ changes for both stages (Figs. 8(c1-c2)), which are calculated by integrating hourly PM$_{2.5}$ changes induced by each process from 00:00LST to 23:00LST (Figs. 8(b1-b2)). In WRF-Chem, DRYD is intermingled with vertical diffusion, so changes in the column burden during vertical mixing can be attributed to DRYD (Tao et al., 2015). Following Tao et al. (2015), we define vertical mixing (VMIX) as the sum of DIFF and DRYD. As shown in Figs. 8(c1-c2), contributions of AERC and VMIX process to 24-h PM$_{2.5}$ changes were +43.8 (+17.9) μg m$^{-3}$ and -161.6 (-221.6) μg m$^{-3}$ for Stage_1 (Stage_2), respectively. Small differences were found for contributions from other processes between Stage_1 and Stage_2 (differences smaller than 10 μg m$^{-3}$). Therefore, the PM$_{2.5}$ increase over the BTH region during haze formation stage (Stage_1) was mainly attributed to strong production by aerosol chemistry process and weak removal by vertical mixing process. On the contrary, during haze elimination stage (Stage_2), more aerosols in BTH were dispersed to the upper atmosphere or subsided to the ground. What's more, the dry cold air from the north decreased the specific humidity (as shown in Fig. 7(b)) in BTH, leading to weaker production of secondary aerosols by aerosol chemistry process.



## 5 Aerosol radiative effects (ARE) on the haze episode

Suspended aerosol particulates can perturb the earth-atmosphere radiation balance, alter meteorological fields, and further affect air quality (Wang et al., 2017a). Previous studies have demonstrated the significance of aerosol-radiation feedbacks on air quality in BTH, especially during winter haze periods. The aerosol radiative forcing was reported to increase by 11.9%-28.7% of the near-surface $PM_{2.5}$ concentrations (Gao et al., 2015; Gao et al., 2016; Qiu et al., 2017; Zhou et al., 2018). However, the detailed influence mechanisms (i.e., the prominent physical or chemical processes responsible for the aerosol radiative impacts on $PM_{2.5}$ concentrations) are still unclear. In this section, we examine the effects of aerosol radiative forcing on meteorological parameters and $PM_{2.5}$ levels during the haze episode, with a special focus on the detailed influence mechanism by using the IPR analysis.

### 5.1 Effects of aerosol radiative forcing on meteorological parameters and $PM_{2.5}$ concentrations

Figure 9 illustrates the impacts of aerosols on the downward shortwave radiative flux (SW) at the surface (BOT_SW) and in the atmosphere (ATM_SW), calculated by subtracting the model results of NoARE from those of CTL, during Stage_1, Stage_2, and the whole simulation period. Downward SW at the surface was strongly decreased when ARE was considered, especially over high aerosol-loading regions during heavily polluted periods. It was known that aerosols could scatter and absorb incoming solar radiation and lead to surface dimming. Besides, in-cloud particles could change the lifetime and albedo of cloud and influence the shortwave radiation at the ground. Generally, the shortwave radiation fluxes at the surface averaged over BTH were reduced by 36.5% (31.6 W m$^{-2}$) in Stage_1, 18.3% (16.6 W m$^{-2}$) in Stage_2, and 24.1% (21.5 W m$^{-2}$) during the whole simulation period, respectively. Contrary to the significant negative effects at the surface, as a result of ARE, the downward SW fluxes in the atmosphere averaged over BTH were increased by 84.7% (25.5 W m$^{-2}$) in Stage_1, 37.4% (10.8 W m$^{-2}$) in Stage_2, and 53.9% (15.7 W m$^{-2}$) during the whole period, respectively.

The impacts of ARE (including aerosol direct and indirect effects) on meteorological parameters and $PM_{2.5}$ concentrations are analyzed in Fig. 10. Because less SW could reach the ground, near-surface temperature was decreased over BTH (Fig. 10(a)), especially during stage_1 when $PM_{2.5}$ concentrations were higher, and the largest decrease was up to 2 k. Meanwhile, the increased SW in the atmosphere could warm the upper air. As a result, a more stable atmosphere was



expected. It is known that the atmosphere stability can be exactly characterized by the profile of equivalent potential

temperature (EPT) (Bolton, 1980; Zhao et al., 2013; Yang et al., 2016). If EPT rises with height, the atmosphere is stable. As

shown in Fig. 10(b), the EPT was decreased in the lower atmosphere (below 1000 m) with the largest decrease of 3 k on

December 22, but increased in the upper atmosphere (above 1200 m). The change in the EPT profile indicated that ARE

could lead to a more stable atmosphere, which further weakened vertical movement in BTH (Fig. 10(c)). As a result of ARE,

the PBLH was decreased and the relative humidity in the lower atmosphere was increased (Fig. 10(d)). All the changes in

meteorological variables were beneficial for $PM_{2.5}$ accumulation in the lower atmosphere (Fig. 10(e)). The daily maximum

increase of $PM_{2.5}$ concentration was 43.2 µg m$^{-3}$ due to ARE. It was noticed that ARE had a negative impact on the

near-surface $PM_{2.5}$ concentrations during December 23-24, which could be explained that absorbing aerosols (i.e., BC)

induced anomalous northeasterlies, and the relatively clean air transported from the northeastern regions to BTH (Fig. S4).

### 5.2 Influence mechanism of aerosol radiative effects

Since variations in $PM_{2.5}$ concentrations are directly caused by physical and chemical processes (Zhu et al., 2015), the

IPR method is then used to investigate the detailed influence mechanism (i.e., the prominent physical or chemical processes

responsible for the aerosol radiative impacts on haze episodes). Figs. 11(a-b) show the diurnal variations of $PM_{2.5}$

concentrations in NoARE and CTL cases averaged over the BTH region in Stage_1. A 24-h increase of 40.8 µg m$^{-3}$ was

simulated in NoARE case. When aerosol radiative forcing was considered, the 24-h increase of $PM_{2.5}$ concentration was 50.4

µg m$^{-3}$. The enhancement of 9.6 µg m$^{-3}$ (23.5%) induced by ARE could be mainly attributed to the contributions of VMIX,

TRAN, and AERC processes, as shown in Fig. 11(c). The vertical mixing was strongly restrained by ARE, therefore fewer

particles diffuse from the surface to the upper atmosphere, resulting in the accumulation of $PM_{2.5}$ in a lower atmospheric

boundary layer. The changes induced by ARE in contributions of VMIX process exhibited positive values in the lower layers

and negative values in the upper layers (Fig. S5(a)). Generally, the VMIX process contributed +39.8 µg m$^{-3}$ to the

enhancement in 24-h $PM_{2.5}$ increase (+9.6 µg m$^{-3}$) for Stage_1. The TRAN process, however, contributed -38.6 µg m$^{-3}$.

Constrained vertical mixing due to ARE could increase aerosol precursors and water vapor in the thin boundary layer, and

enhance the formation of secondary particles. Generally, the AERC process contributed +5.1 µg m$^{-3}$. The positive

contributions of AERC process were mainly distributed over the high polluted regions in BTH (Fig. S5(b)). Detailedly, the

changes in $SO_4^{2-}$, $NO_3^-$, and $NH_4^+$ concentrations during the daytime from 10:00 to 16:00LST were -0.5 μg m$^{-3}$, +5.9 μg

m$^{-3}$, and +2.9 μg m$^{-3}$, respectively. The decreased near-surface temperature caused by ARE may suppress the chemical

formation of $SO_4^{2-}$. Generally, the total contribution of VMIX, TRAN, and AERC processes to the change in 24-h PM$_{2.5}$

increase caused by ARE was +6.3 μg m$^{-3}$, and the restrained vertical mixing could be the primary reason for near-surface

PM$_{2.5}$ increase when aerosol radiative forcing was considered.

      Fig. 12(a) shows the vertical profiles of the 24-h increases in PM$_{2.5}$ concentrations (23:00LST minus 00:00LST)

averaged over BTH during Stage_1 in CTL and NoARE cases. Below ~400 m (between L01 and L04), the 24-h increase

simulated by CTL was larger than that in NoARE, which could be explained by that the positive contributions of VMIX and

AERC exceeded the negative contributions of TRAN in the lower atmosphere when aerosol radiative effect was considered

(Fig. 12(b)). However, in the upper layers (400 to 2000 m or L05 to L15), aerosol radiative forcing weakened the 24-h PM$_{2.5}$

increase during Stage_1. When aerosol radiative effect was considered, fewer particulate matters, precursors and water vapor

were diffused from the surface to the upper layers, and therefore fewer particles were formed in the upper layers. Despite of

the positive contributions of TRAN, the net contributions of VMIX, AERC, and TRAN to PM$_{2.5}$ changes caused by ARE in

the upper atmosphere were negative.

## 6. Conclusions and discussions

      In this study, an online coupled mesoscale meteorology–chemistry model (WRF–Chem) with an improved integrated

process rate (IPR) analysis (i.e., process analysis) scheme is applied to investigate the formation and evolution mechanisms

of a severe haze episode happened in the BTH region during 20–29 December 2015. Sensitivity experiments are conducted

to examine the contributions of local emission and outside transport to the absolute PM$_{2.5}$ concentrations during the haze

episode, while the IPR analysis is used to quantify the contributions of each detailed physical/chemical process to the

variations in the PM$_{2.5}$ concentration. The impacts of aerosol radiative forcing, including direct and indirect effects, on

meteorological parameters and PM$_{2.5}$ levels during the haze episode are also quantified, with a special focus on the detailed

influence mechanism (i.e., the prominent physical or chemical processes responsible for the aerosol radiative impacts on the



haze episode).

The measurements from NOAA and WRMC–BSRN are used to evaluate the simulated meteorological parameters; the observations from CNEMC are used to evaluate the simulated $PM_{2.5}$ concentrations. Generally, good agreements between observations and simulations are achieved for both meteorological and chemical variables, indicating that the WRF–Chem model has the capability to reproduce the haze episode.

Spatial–temporal evolutions of surface–layer $PM_{2.5}$ concentrations, and the contributions of local emission and outside transport to the absolute $PM_{2.5}$ concentrations, were firstly analyzed. During the aerosol accumulation stage (December 20–22, Stage_1), the near-surface $PM_{2.5}$ concentrations in BTH experienced a consistent increase, with the average $PM_{2.5}$ concentration reaching 250.0 μg m$^{-3}$, far beyond the threshold value of "heavily polluted". The contributions of local emission and regional transport to the $PM_{2.5}$ concentrations averaged over BTH were comparable (42.3% and 36.6%, respectively), meaning the combined effect together resulted in the high $PM_{2.5}$ concentrations. During the aerosol dispersion stage (December 23–27, Stage_2), the near–surface $PM_{2.5}$ concentrations in BTH underwent a consistent decrease, and the average $PM_{2.5}$ concentration was 107.9 μg m$^{-3}$. The contributions of local emission and regional transport were 50.9% and 24.3%, respectively. Therefore, the relatively high $PM_{2.5}$ concentration during Stage_2 was principally caused by local emission. During December 28–29 (Stage_3), another haze event was formed and developed.

The IPR analysis was then used to explain the reason for $PM_{2.5}$ increase during Stage_1 and $PM_{2.5}$ decrease during Stage_2, by quantifying the contributions of each physical/chemical process to variations in $PM_{2.5}$ concentrations. During both stages, the dominant sources of surface–layer $PM_{2.5}$ were emission (EMIS) and aerosol chemistry (AERC) process, while the main sinks were turbulent diffusion (DIFF), advection (TRAN), and dry deposition (DRYD) process. The $PM_{2.5}$ concentration increased by 50.4 μg m$^{-3}$ (23:00LST minus 00:00LST) during Stage_1 but decreased by 41.5 μg m$^{-3}$ during Stage_2. Contributions of AERC and VMIX (vertical mixing, the sum of DRYD and DIFF) process to 24–h $PM_{2.5}$ changes were +43.8 (+17.9) μg m$^{-3}$ and −161.6 (−221.6) μg m$^{-3}$ for Stage_1 (Stage_2), respectively. Small differences in contributions from other processes were found between Stage_1 and Stage_2. Therefore, the $PM_{2.5}$ increase over BTH during haze formation stage (Stage_1) was attributed to strong production by aerosol chemistry process and weak removal by vertical mixing process.

Aerosol radiative effects (ARE) on the meteorological parameters and $PM_{2.5}$ concentrations were also examined. When aerosol radiative forcing was considered, the downward shortwave flux was decreased at the surface but increased in the atmosphere. The equivalent potential temperature was decreased in the lower atmosphere but increased in the upper atmosphere, leading to a more stable atmosphere. As a result of ARE, the PBLH was decreased and the relative humidity in the lower atmosphere was increased, which were beneficial for $PM_{2.5}$ accumulation in the lower atmosphere. The daily maximum increase of the near–surface $PM_{2.5}$ concentration was 43.2 μg m$^{-3}$.

Finally, the IPR method was used to investigate the detailed influence mechanism of aerosol radiative impacts. When aerosol radiative feedback was considered, the 24–h $PM_{2.5}$ increase was enhanced by 9.6 μg m$^{-3}$ (23.5%) during Stage_1, which could be mainly attributed to the contributions of VMIX, TRAN, and AERC processes. Generally, the VMIX, TRAN, and AERC processes contributed +39.8 μg m$^{-3}$, −38.6 μg m$^{-3}$, and +5.1 μg m$^{-3}$ to the enhancement in 24–h $PM_{2.5}$ increase (+9.6 μg m$^{-3}$), respectively. The restrained vertical mixing could be the primary reason for near-surface $PM_{2.5}$ increase when aerosol radiative forcing was considered.

There are some limitations in this work. The uncertainty of the MIX anthropogenic emission inventory, the lack of secondary organic aerosols, and the missing mechanisms of some heterogeneous reactions may lead to the uncertainties in the final simulation results. Besides, conclusions drew from a case study may not represent a full view of the underlying mechanisms of haze formation and elimination. Better understanding will be attained by conducting multiple case simulations in the future. What's more, an anomalous northeasterly induced by absorbing aerosols was observed, leading to a decrease in the near-surface $PM_{2.5}$ concentrations during December 23–24 2015 in BTH, which was different from previous studies that reported light-absorbing aerosols could worsen air quality (Li et al., 2016; Huang et al., 2018; Gao et al., 2018). More experiments should be designed to examine the changes in atmospheric thermal and atmospheric dynamic caused by absorbing aerosol radiative forcing and their effects on haze episodes in the future.

**Data availability**

Observational datasets and simulation results are available upon request to the corresponding author (hongliao@nuist.edu.cn).

**Author contributions**

HL and LC conceived the study and designed the experiments. LC and JZ performed the simulations and carried out the data analysis. YG, MZ, YQ and NL provided useful comments on the paper. LC prepared the manuscript with contributions from all co-authors.

**Competing interests**

The authors declare that they have no conflict of interest.

**Special issue statement**

This study is part of the special issue "Regional transport and transformation of air pollution in eastern China". It is not associated with a conference.

**Acknowledgements**

This study was supported by the National Natural Science Foundation of China (91744311), the University Natural Science

Research Foundation of Jiangsu Province (18KJB170012), and the Startup Foundation for Introducing Talent of NUIST (2018r007).



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



**Table 1. Parameterizations used in the WRF-Chem model**

| Options | WRF-Chem |
|---|---|
| Microphysics option | Purdue Lin scheme |
| Longwave radiation option | RRTMG scheme |
| Shortwave radiation option | RRTMG scheme |
| Surface layer option | Revised MM5 Monin-Obukhov scheme |
| Land surface option | Unified Noah land-surface model |
| Urban canopy model | Single-layer UCM scheme |
| Boundary layer option | YSU scheme |
| Cumulus option | Grell 3D ensemble scheme |
| Photolysis scheme | Fast-J |
| Dust scheme | Shao_2004 |
| Chemistry option | CBMZ |
| Aerosol option | MOSAIC |
| Analysis nudging | On |



**Table 2. Experimental design**

| Case Description | Anthropogenic Emission | Aerosol Direct Effect | Aerosol Indirect Effect |
|---|---|---|---|
| CTL | Y | Y | Y |
| NoAnth | Without emission in the whole domain | Y | Y |
| NoBTH_Anth | Without emission in BTH | Y | Y |
| OnlyBTH_Anth | Only emission in BTH | Y | Y |
| NoARE | Y | N | N |

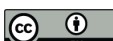



**Table 3. Statistical metrics between observations and simulations**

| Variables | nstd[b] | OBS[b] | SIM[b] | MB[b] | GE[b] | NMB[b] | RMSE[b] | MFB[b] | MFE[b] | IOA[b] | R[b] |
|---|---|---|---|---|---|---|---|---|---|---|
| $T_2$ (k)[a] | 12 | 271.0 | 272.0 | 1.1 | 2.1 | 0.4 | 2.6 | 0.4 | 0.8 | 0.9 | 0.9 |
| $RH_2$ (%)[a] | 12 | 69.6 | 59.6 | -10.0 | 14.0 | -14.3 | 18.1 | -15.2 | 22.6 | 0.7 | 0.8 |
| $WS_{10}$ (m s$^{-1}$)[a] | 12 | 2.4 | 3.1 | 0.7 | 1.4 | 29.1 | 1.8 | 33.3 | 58.0 | 0.7 | 0.8 |
| $WD_{10}$ (°)[a] | 12 | 181.7 | 179.4 | -2.3 | 89.4 | -1.3 | 135.6 | -4.6 | 59.6 | 0.3 | 0.6 |
| $PM_{2.5}$ (µg m$^{-3}$) | 59 | 210.0 | 194.3 | -15.7 | 79.2 | -7.5 | 110.0 | 2.8 | 44.3 | 0.7 | 0.8 |

[a]$T_2$: temperature at 2 m (k); $RH_2$: relative humidity at 2 m (%); $WS_{10}$: wind speed at 10 m (m s$^{-1}$); $WD_{10}$: wind direction at 10 m (°).

[b]nstd: the number of observation sites; OBS: the average observations; SIM: the average simulations; MB: mean bias; GE: gross error;

NMB: normalized mean bias (%); RMSE: root mean square error; MFB: mean fractional bias (%); MFE: mean fractional error (%); IOA:

5    index of agreement; R: correlation coefficient.



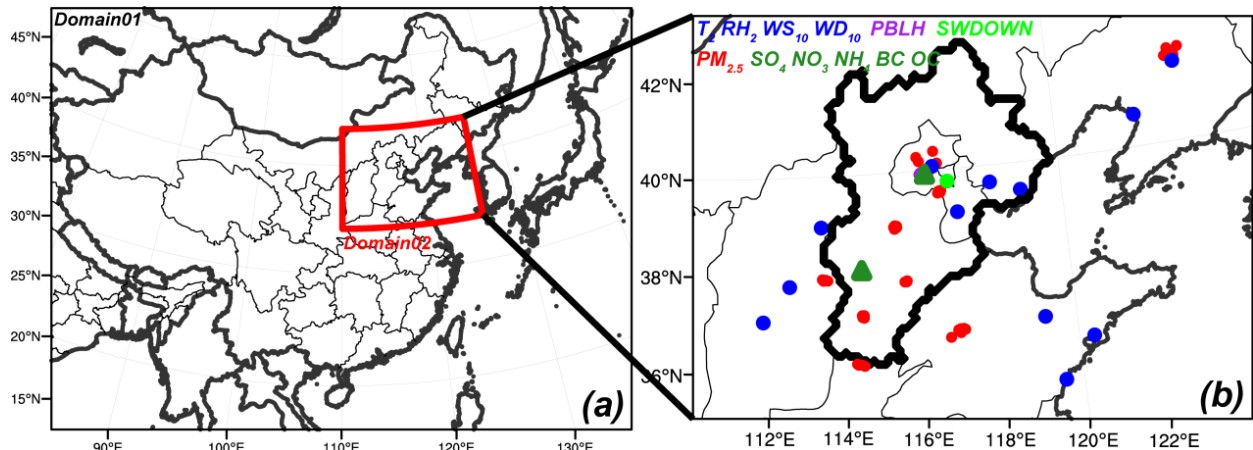

**Figure 1. (a) Map of the two nested model domains. (b) Locations of the observations used for model evaluation.**

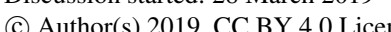



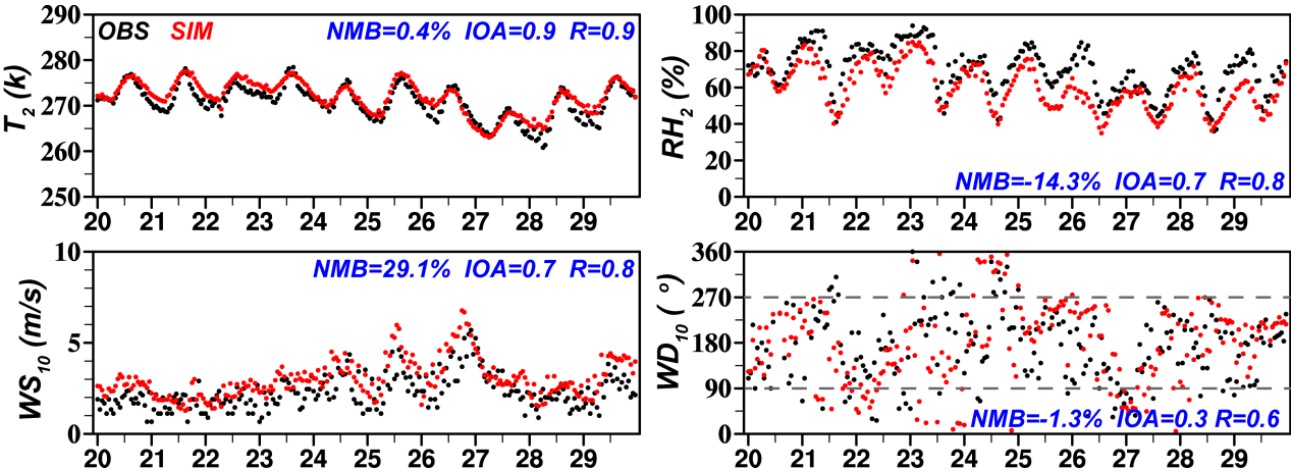

**Figure 2. Time series of observed (shown in black dots) and simulated (shown in red dots) hourly 2 m temperature (T$_2$, k), 2 m relative humidity (RH$_2$, %), 10 m wind speed (WS$_{10}$, m s$^{-1}$), and 10 m wind direction (WD$_{10}$, ) averaged over the 12 stations during 20–29 December 2015.**



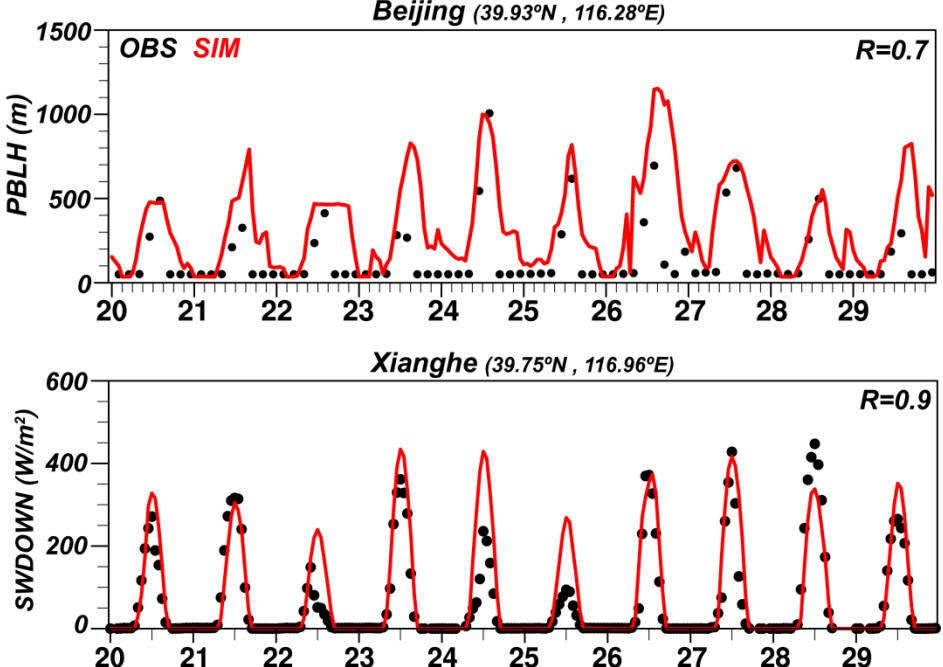

**Figure 3. Time series of observed (shown in black dots) and simulated (shown in red lines) hourly planetary boundary layer height**

**(PBLH, m) in Beijing (39.93 N, 116.28 E) and shortwave downward radiation flux (SWDOWN, W m$^{-2}$) at the Xianghe Station**

5    **(39.75 N, 116.96 E) from 20 to 29 December 2015. Notably, PBLH provided by GDAS of NOAA are in 3-hour intervals.**



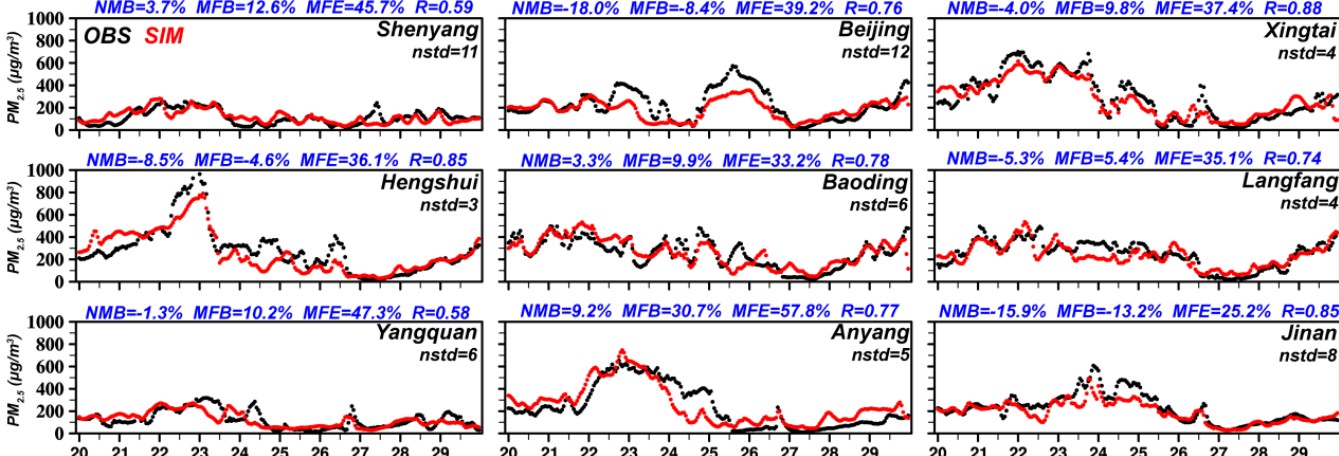

**Figure 4. Time series of observed (shown in black dots) and simulated (shown in red dots) hourly PM$_{2.5}$ concentrations (μg m$^{-3}$) in the nine cities (Shengyang, Beijing, Xingtai, Hengshui, Baoding, Langfang, Yangquan, Anyang, and Jinan) from 20 to 29 December 2015. The nstd in each panel represents the number of observation sites in each city.**





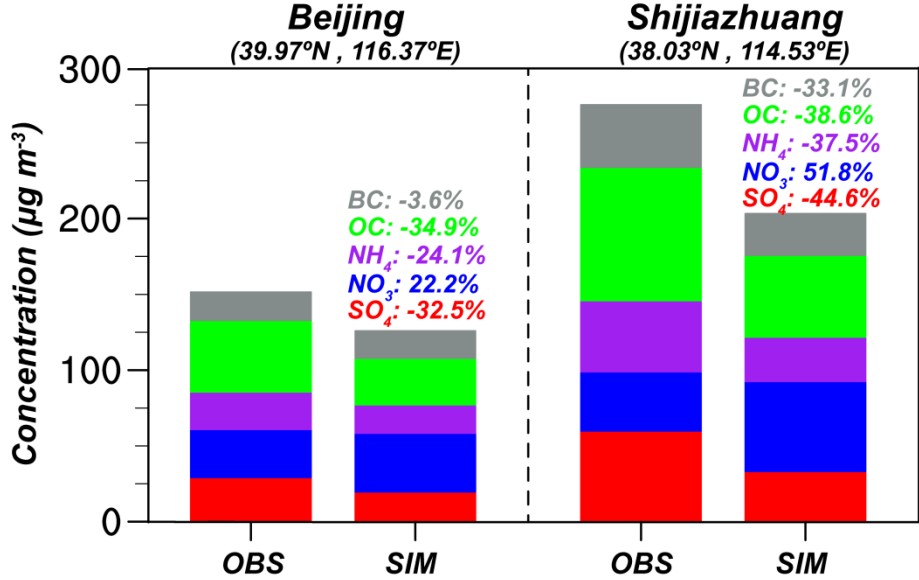

**Figure 5.** Comparison of observed and simulated surface–layer mass concentrations (μg m$^{-3}$) of $SO_4^{2-}$ (red), $NO_3^-$ (blue), $NH_4^+$ (purple), OC (green), and BC (gray) in (a) Beijing (39.97 N, 116.37 E) and (b) Shijiazhuang (38.03 N, 114.53 E) averaged over 20–29 December 2015. Also listed in colored numbers are normalized mean biases (NMBs) for each species.



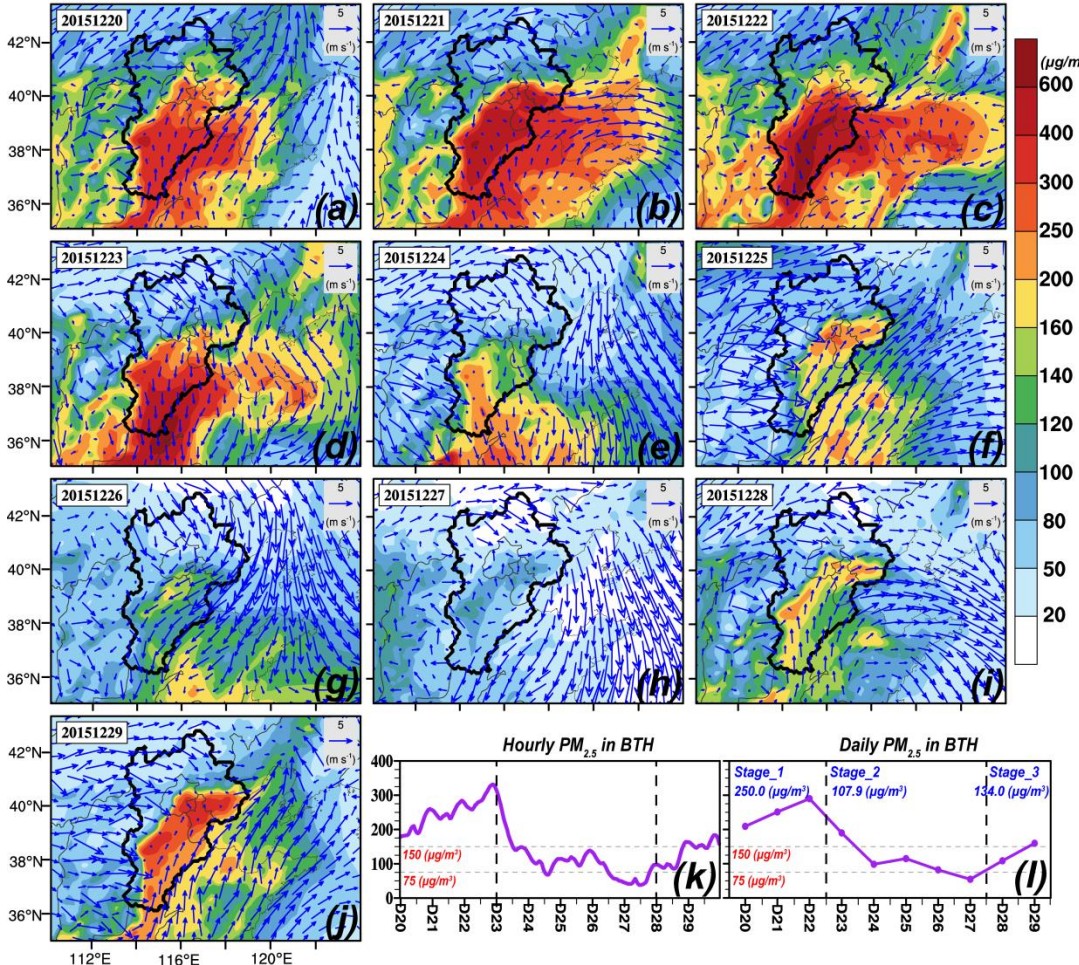

**Figure 6.** (a-j) Spatial distributions of simulated daily $PM_{2.5}$ concentrations (shaded, µg m$^{-3}$) and wind vectors (arrows, m s$^{-1}$) from 20 to 29 December 2015. Time series of simulated hourly and daily $PM_{2.5}$ concentrations averaged over the BTH region are also shown in (k) and (l), respectively.



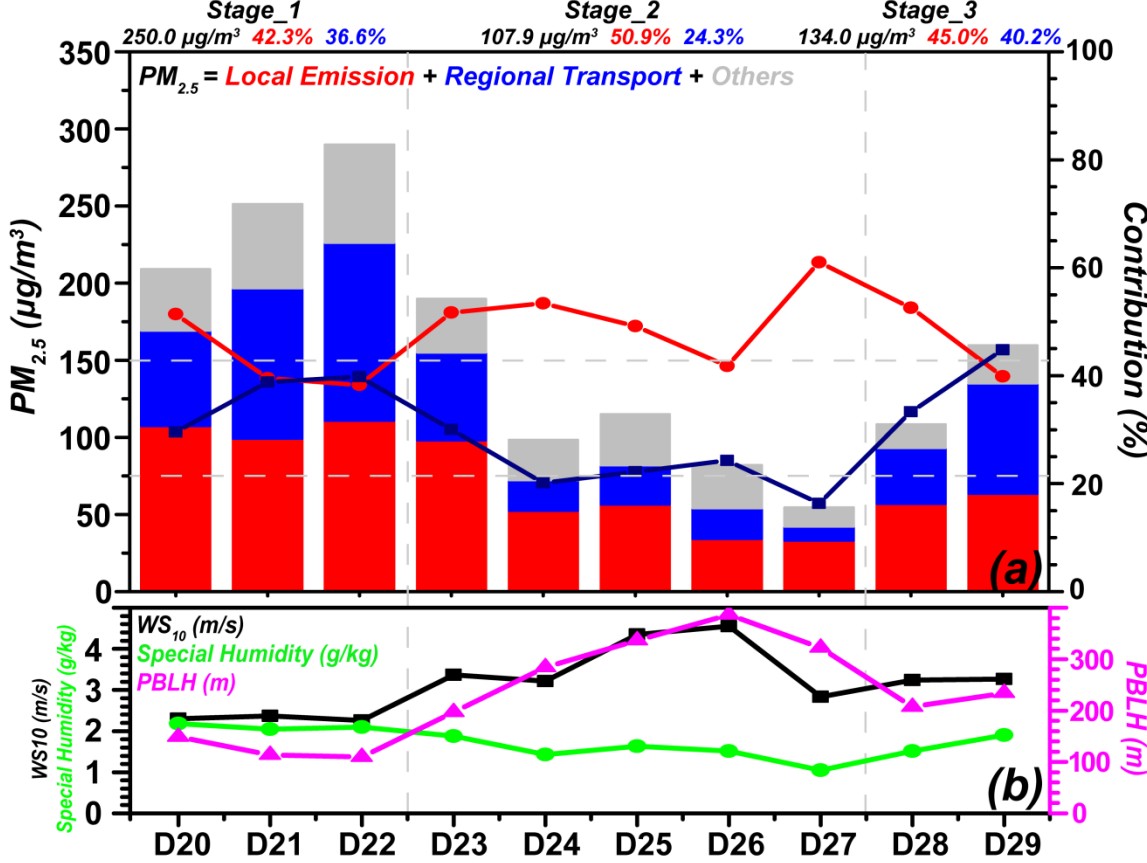

**Figure 7. (a)** Contributions of local emission (shown in red) and regional transport (shown in blue) to the near-surface $PM_{2.5}$ concentrations averaged over the BTH region from 20 to 29 December 2015. The absolute contributions are shown in bars ($\mu g\ m^{-3}$) and the percentage contributions are shown in lines (%). The $PM_{2.5}$ concentration and the percentage contributions averaged over each stage are listed at the top of (a). Simulated daily 10 m wind speed ($WS_{10}$, $m\ s^{-1}$, shown in black dot line), special humidity (g $kg^{-1}$, shown in green dot line), and PBLH (m, shown in purple dot line) averaged over BTH are also shown in (b).



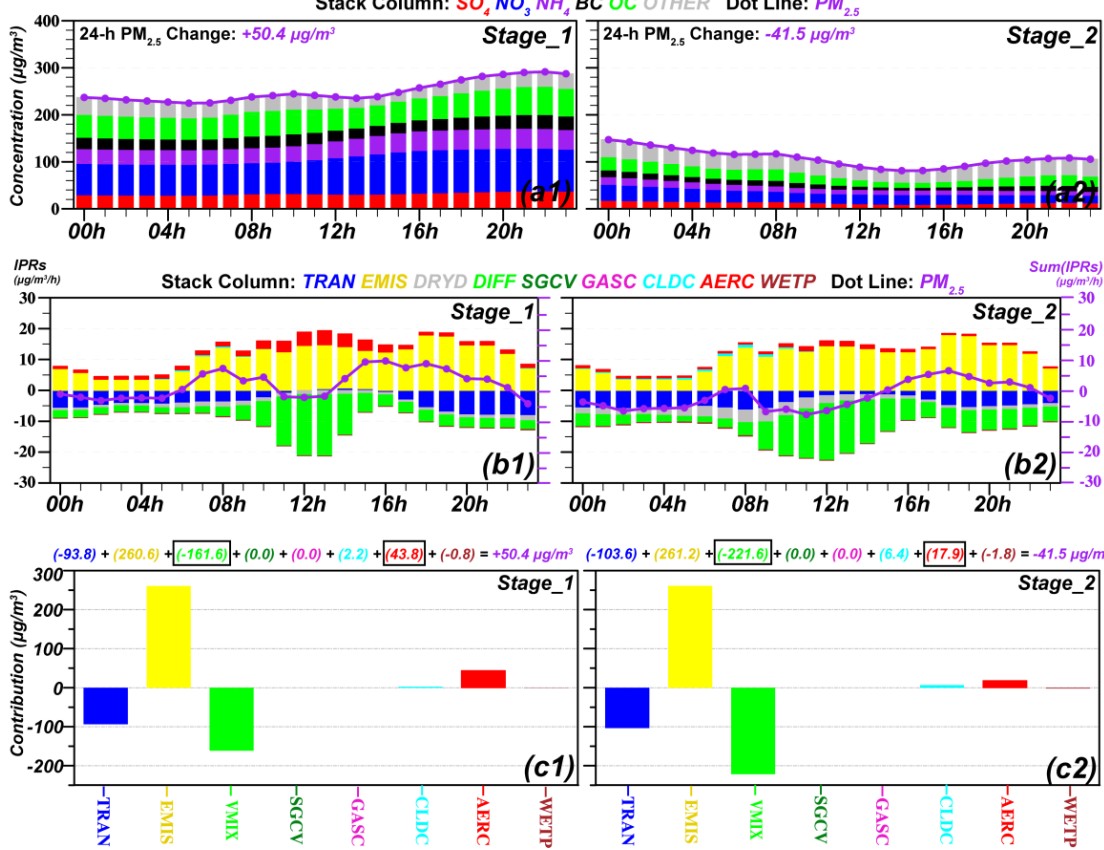

**Figure 8.** (a1-a2) Diurnal variations of PM$_{2.5}$ concentrations averaged over BTH during Stage_1 and Stage_2 (shown by purple dot lines). The colored bars represent different components. Also shown at the top left corner of each panel is the 24-h change in PM$_{2.5}$ concentration (23:00LST minus 00:00LST). (b1-b2) The hourly PM$_{2.5}$ changes induced by each physical/chemical process by using the IPR analysis method (shown by colored bars). The purple dot lines represent hourly PM$_{2.5}$ changes induced by all processes, also indicating the differences between current and previous-hour PM$_{2.5}$ concentrations. (c1-c2) Contributions of each physical/chemical process to 24-h PM$_{2.5}$ changes.



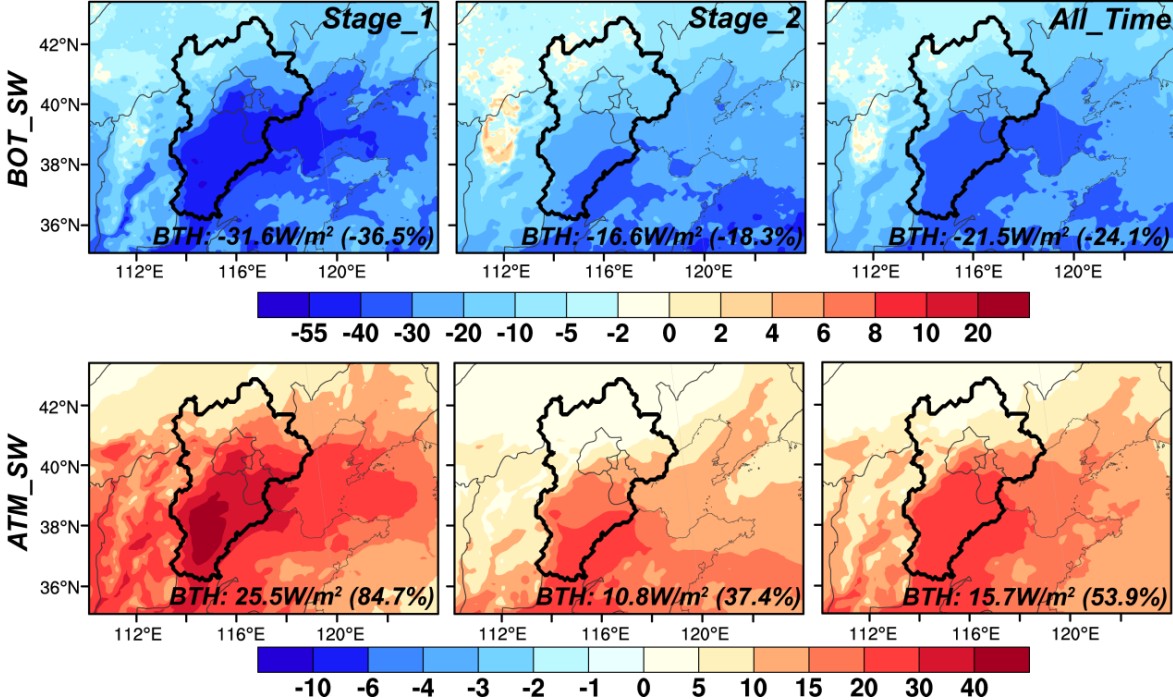

**Figure 9. The differences in simulated all-sky radiative forcing (W m$^{-2}$) between CTL and NoARE cases (CTL minus NoARE) averaged over Stage_1, Stage_2, and the whole simulation period. "BOT_SW" and "ATM_SW" denote the downward shortwave radiative flux at the surface and in the atmosphere, respectively. The calculated differences in the simulated radiative forcing averaged over BTH for each stage are also shown at the bottom of each panel.**




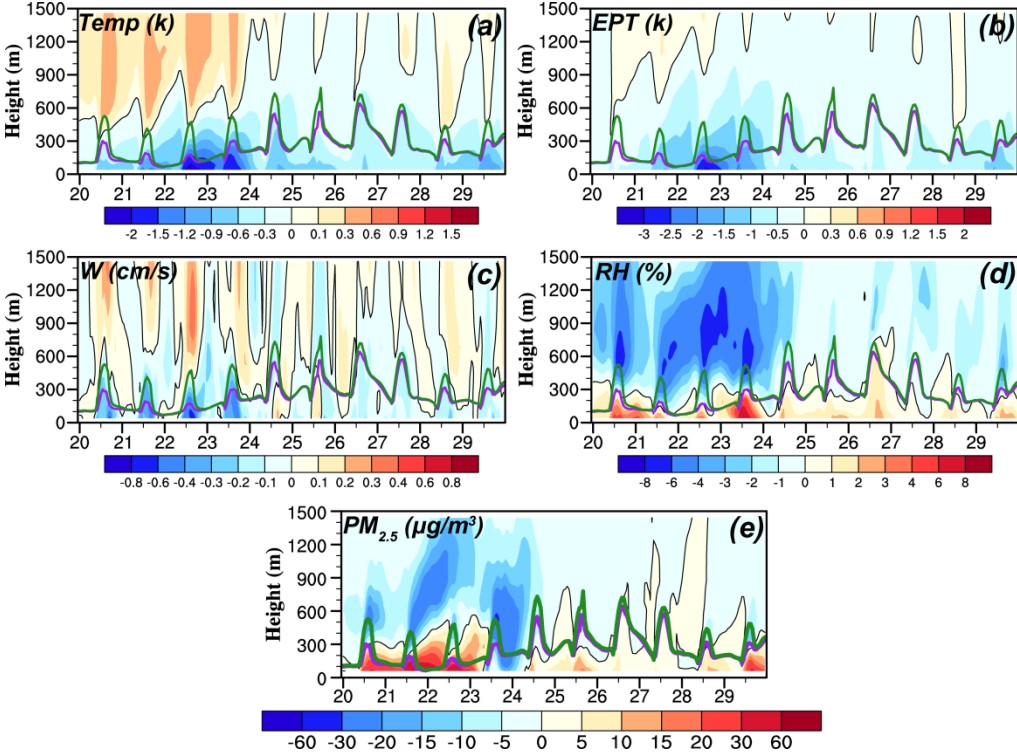

**Figure 10. Time series of differences in (a) temperature (k), (b) equivalent potential temperature (k), (c) vertical wind speed (cm s$^{-1}$), (d) relative humidity (%), and (e) PM$_{2.5}$ concentration (µg m$^{-3}$) between CTL and NoARE cases (CTL minus NoARE) averaged over the BTH region. The purple and green lines denote the simulated PBLH in CTL and NoARE cases, respectively.**

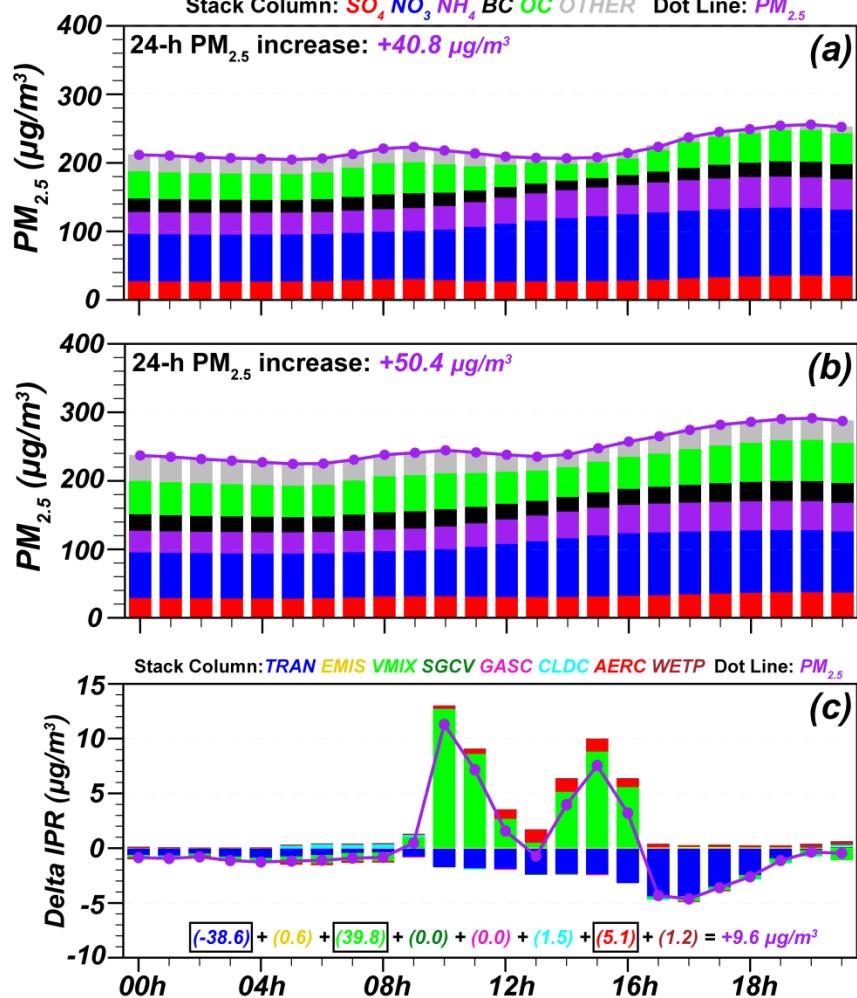

**Figure 11. Diurnal variations of the near-surface PM$_{2.5}$ concentrations in (a) NoARE and (b) CTL simulations averaged over the BTH region during Stage_1 (shown by purple dot lines). The colored bars represent different components. Also shown at the top left corner of each panel is the 24-h increase in PM$_{2.5}$ concentration (23:00LST minus 00:00LST). (c) Differences in hourly IPRs caused by aerosol radiative forcing (CTL minus NoARE). The numbers listed in (c) represent the contributions of each process to the change in 24-h PM$_{2.5}$ increase caused by aerosol radiative forcing.**



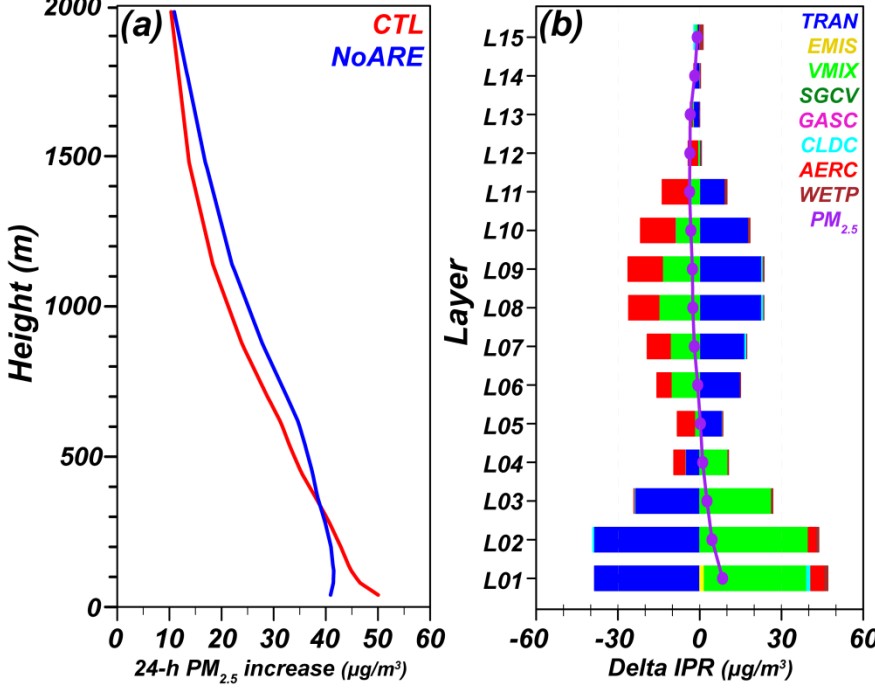

**Figure 12. (a) Vertical profiles of the 24-h increases in PM$_{2.5}$ concentrations (23:00LST minus 00:00LST) averaged over BTH during Stage_1 in CTL and NoARE cases. (b) Vertical profiles of the differences in the 24-h PM$_{2.5}$ increases caused by aerosol radiative effect (CTL minus NoARE, as show in purple dot line), and the contributions of each physical/chemical process (as shown in colored bars).**