# Peer review of "Assessing the formation and evolution mechanisms of severe haze pollution in Beijing-Tianjin-Hebei region by using process analysis"

_Atmospheric Chemistry and Physics, 2019_

## Referee Comment (RC1) · Anonymous Referee #1 · 20 Apr 2019

In this study, the authors examined the formation and evolution mechanisms of a haze event happened over Beijing–Tianjin–Hebei (BTH) in December 2015 using WRF-Chem model with a newly developed integrated process rate (IPR) analysis technique. They found that the $PM_{2.5}$ increase during aerosol accumulation stage was mainly attributed to strong production by aerosol chemistry process and weak removal by vertical mixing process, and the restrained vertical mixing could be the primary reason for near-surface $PM_{2.5}$ increase when aerosol radiative feedback was considered. This study is interesting, and the results are solid. IPR technique provides a fundamental information of the physical/chemical processes of aerosol change. The manuscript is well written. I would suggest publication after addressing my comments below.

Through IPR analysis, the authors found that $PM_{2.5}$ increase in the stage_1 was due to increased aerosol chemical production and decreased vertical mixing removal, but they did not explain why aerosol chemical production increased/vertical mixing removal decreased during this period. In Section 4.1, the authors used changes in synoptic conditions and atmospheric circulation to explain the aerosol variation, which seems not consistent with the IPR results. The specific humidity did not have a visible change during Dec.20-22, so the aerosol chemical production was not due to the humidity. Probably it was due to increases in aerosol precursors from regional transport.

Although the model reproduced well the $PM_{2.5}$ variation during the haze event, it strongly underestimated concentrations of most aerosols, but overestimated nitrate concentration (Figure 5). However, during stage_1, it looks that most PM2.5 increase is due to nitrate increase (Figure 8a). The authors need to discuss the potential influences of model biases on the results.

Minor comments:
Page 1 Line 23: Delete 'happened'.

Page 1 Line 24 and following parts: Please change 'outside transport' to 'regional transport'.

Page 3 Line 12: Delete 'even'.

Page 4 Line 5: Change 'contribution' to 'contributor'.

Page 4 Line 14: Recent studies found that black carbon-East Asian winter monsoon interactions and dust-wind interactions can also intensify winter haze in eastern China (e.g., Yang et al., 2017; Lou et al., 2019). The authors may would like to cite these studies.

Page 8 Line 19: What does emission source mean and how it affects aerosol variation with 24-hr (Figure 8). Does the model include diurnal variation of emission? Why contributions of EMIS are different between stage_1 and stage_2?

Page 9 Line 19: Change 'closing' to 'turning off'.

Page 11 Line 4: How these matrices calculated?

Page 14 Line 14-22: Please rephrase this paragraph by illustrating absolute change first then percentage change (relative to what?). And what are the rest of contribution, from natural emission or emission outside the domain?

Page 15 Line 7: Please clarify that the dominant sources of surface-layer PM2.5 'variation'.

Figure 7: Change 'special' to 'specific' in caption and figure.

Figure 11: Change '18h' to '20h'

References:
Yang, Y., L. M. Russell, S. Lou, H. Liao, J. Guo, Y. Liu, B. Singh, and S. J. Ghan, Dust-wind interactions can intensify aerosol pollution over eastern China, Nat. Commun., 8, 15333, doi:10.1038/ncomms15333, 2017.

Lou, S., Y. Yang, H. Wang, S. J. Smith, Y. Qian, and P. J. Rasch, Black carbon amplifies haze over the North China Plain by weakening the East Asian winter monsoon, Geophys. Res. Lett., 46, 452–460, doi:10.1029/2018GL080941, 2019.

---

## Referee Comment (RC2) · Anonymous Referee #2 · 21 Apr 2019

Review of: "Assessing the formation and evolution mechanisms of severe haze pollution in Beijing−Tianjin−Hebei region by using process analysis" by Chen et al.

The authors quantitatively examined the cause of a severe haze event over Beijing–Tianjin–Hebei (BTH) through decomposing contributions from emissions, physical and chemical processes, using the WRF–Chem model equipped with an improved integrated process rate (IPR) analysis scheme. The IPR scheme also reveals the dominant role of aerosol radiative effects in haze formation is physical rather than chemical. Such an IPR scheme merits application in future relevant studies. Overall, the manuscript is well structured and should be of great interest to ACP readers. I recommend it is

publishable after my minor comments/suggestions are addressed.

General comments: -Local vs. regional contributions. From the sensitivity simulations, local emissions and regional transport account for 80% of total PM2.5. What does the contribution from "others" (Fig.7) mean? is it because the non-linear chemical formation for secondary aerosol or contribution from aerosol precursors outside of BTH?

-Advection contribution. A negative value for advection is diagnosed by the IPR scheme. From my understanding the advection means horizontal transport, which should not be always a negative contribution, and instead it may contribute a lot to PM2.5 if taking the value of regional contribution as an equivalent.

-Aerosol radiative effects. It is considerate to include the aerosol indirect effect, though this process contributes marginally in less-cloud wintertime. But the authors failed to show/discuss how aerosol indirect effect is expressed from IPR result. For example, in Fig. 11 the CLDC (1.5 $\mu$g m-3) and WETP (1.2 $\mu$g m-3) can be taken as the result of aerosol indirect effect. It needs to clarify.

Specific comments:

-The presentation of this work would be greater if some editorial aspects are improved.

Use of %: P1L27. Here and elsewhere in the text, please round off 250% instead of 250.0%. The decimal doesn't make sense in terms of model bias.

Please check the appropriate usage of hyphen (-) (e.g., near-surface) and en-dash (–) (e.g., December 20–22).

Fig.1: The purple dot (?) for PBLH is hardly to see. Reduce the size of green triangle or increase its transparency.

Fig.4: I suggest the use of NMB and correlation coefficient are good enough for model evaluation. Reader gets lost in so many numbers.

Fig.6: I can't see any different for (k) and (i), and there is also no discussion in the text.

Remove one of them.

Fig.8: Move the middle panel (b1 and b2) towards right. Is there any difference in Y-axis of (b1) and (b2). I suppose they are the same.

-P1L24: there is any special meaning for "absolute" PM2.5. If not, please "absolute" when it is unnecessary.

-P2L6 & P5L8: remove "and so on".

-P3L7: I don't think severe haze frequently occurs in wintertime over PRD region, and neither of your two references support this.

-P310-12: Health threat by PM2.5 is the most important thing people care about.

-P5L1: remove "(SPM)". You don't use it in the following text.

-P5L3: haze is not actually caused by "the interactions between...". It's a synergy effect by these factors.

-P5L12-13: is there any reference saying "substantial efforts since 2009"? Zheng et al. (2018, ACP, Trends in China's anthropogenic emissions since 2010 as the consequence of clean air actions) shows emissions dropped substantially only after 2013.

-P7L2: which year of anthropogenic emission from MIX?

-P8L20: sub-grid convection (SGCV) is found to be zero in the simulation. It can be possibly due to no precipitation in this case. But why gas-phase chemistry is also zero, or what it specifically means? Nitrate formation is dominantly through gas-phase oxidation of NO2+OH.

-P9L14 & L16: You haven't defined what "two stages" are. Maybe don't need to mention this here.

-P10L12-19: please delete unnecessary "marked in" in this paragraph.

-P11L4-L6 & P12L10-11: The NMB (IOA) and R are the only metrics you used in the
text and they are informative enough. I suggest remove other metrics if possible. Or define them in Table 3.

-P11L10: change "options" to "parameterizations"

-P13L24: add "air quality" before "threshold value"

-P14L8: I suggest change "internal" to "dominant" or "leading"

-P16L2-4: please delete "Suspended. . .during winter haze periods". You have done this in the Introduction section.

-P16L15-16: Is this process important in your case? If not, saying this here reads misleading.

-P16L10: I suggest to move Fig. S4 in the main text. It is interesting to show the important role of absorbing aerosol on regional circulation change. This result is consistent with the simulation by Qiu et al. (2017).

-Conclusions and discussions. The "conclusions" part can be shortened and concise, which should make room for more insightful discussion. I came up some ideas. (1) how the IPR scheme can be further improved? (2) the authors could discuss the possible application of IPR scheme in future haze study (both winter and summer), because the work provides a quantitative analysis of how aerosol radiative effects change PM2.5 through physical and chemical pathways.

---

## Referee Comment (RC3) · Anonymous Referee #3 · 4 May 2019

General Comments

This study investigated a haze event over BTH region in December 2015 with the process analysis method. The study examined the mechanisms underlying the event formation and evolution. They found the event was mainly controlled by the change of vertical mixing. In the end, the study also found that the vertical mixing and transport were two main processes that were responsible for the aerosol radiative feedback. The manuscript is well-written. However, the main point of processing analysis alone is not novel at all. Many previous studies have used this method in multiple air quality models, including WRF-Chem. The study only selected one event as the analysis case.

[Figure]

Although the study found that the vertical mixing was the main contributor to the formation and evolution of the event, I didn't find anything new brought to the community. Besides these general comments, I also have some specific comments.

Specific Comments

1. The study selected the event of Dec. 20-29. Based on the result, it seems that the PM2.5 concentration reached 200 ug/m3 on average on Dec. 20. To better show the formation of event, the simulation and analysis should start from the earlier date to demonstrate the concentration rising from a lower level such as 50 ug/m3 as shown in the end of this event.

2. Line 19-21 of page 6, if the FDDA is turned on for the control and noARE experiments, I do expect some aerosol meteorological feedbacks can be diminished by the FDDA. Free runs without FDDA is preferred for studying aerosol feedback.

3. Line 11-12 of page 7, the MOSAIC aerosol mechanism in WRF-Chem has not been coupled with the Shao dust emission scheme, at least in the publicly released version. If it was coupled in this study or any previous studies, please briefly introduce it and cite the related references.

4. Line 18-24 of page 9, this part is confusing. In WRF-Chem, the aerosol-cloud interaction is linked with wet deposition and cloud aqueous chemistry. If aerosol-cloud interaction is turned off in this study, then the control and noARE experiments should use different chemistry mechanisms, i.e., noARE likely used the one without wet deposition and cloud chemistry. Please provide more details about this. If this is the case, the difference between the control and noARE should include not only the aerosol feedback but also the difference in chemical processes. Based on the results, it seems that there were little cloud and precipitation during the period. The major aerosol feedback is from aerosol-radiation interaction, therefore, it makes more sense that in noARE experiment only the aerosol radiative feedback is turned off with aerosol-cloud interaction not touched. Furthermore, we generally do not call aerosol-cloud interaction as aerosol

radiative effects.

5. Line 11 of page 10, please provide the full name of NOAA READY GDAS. In addition, please provide more information about the PBL data from this dataset. Is it retrieval or direct observation? If it is retrieval, what is the method used for the retrieval?

6. In the figures of hourly timeseries such as Fig. 3 or 4, please specify whether it is local time or UTC time? In Fig. 3, is there a low limit from PBL retrieval? It seems the values are limited to 50 m. The same is applied to the simulations. Any specific reason?

7. Fig. 5 shows the aerosol components at the station of Shijiazhuang? Why not show the total PM2.5 surface concentration at this station as a reference? In addition, since the simulation seems capturing the hourly PM2.5 variation well, why not show the hourly component comparison instead of period average only? It would be interesting and provide useful information.

8. What does the black line represent in Fig. 10?

9. In Fig. 11, the process analysis showed the averaged 24-h change of PM2.5 during the period. What does this mean? Why is the averaged 24-h change important? I think that the change through each stage of the event would be more interesting. Please clarify.

10. In Fig. 12b, using height (m) instead of model levels as the y-axis makes more sense.

---

## Author Comment (AC1) · 4 Jul 2019

**Response to Comments of Reviewer #1**

**(comments in *italics*)**

**Manuscript number:** acp-2019-245

**Title:** Assessing the formation and evolution mechanisms of severe haze pollution in Beijing-Tianjin-Hebei region by using process analysis

*In this study, the authors examined the formation and evolution mechanisms of a haze event happened over Beijing–Tianjin–Hebei (BTH) in December 2015 using WRF-Chem model with a newly developed integrated process rate (IPR) analysis technique. They found that the $PM_{2.5}$ increase during aerosol accumulation stage was mainly attributed to strong production by aerosol chemistry process and weak removal by advection and vertical mixing processes, and the restrained vertical mixing could be the primary reason for near-surface $PM_{2.5}$ increase when aerosol radiative feedback was considered. This study is interesting, and the results are solid. IPR technique provides a fundamental information of the physical/chemical processes of aerosol change. The manuscript is well written. I would suggest publication after addressing my comments below.*

**Response:**

Thanks to the reviewer for the valuable comments and suggestions which are very helpful for us to improve our manuscript. We have revised the manuscript carefully, as described in our point-to-point responses to the comments.

**General Comments:**

1. *Through IPR analysis, the authors found that $PM_{2.5}$ increase in the stage_1 was due to the increased aerosol chemical production and the decreased vertical mixing removal, but they did not explain why aerosol chemical production increased/vertical mixing removal decreased during this period.*

**Response to why aerosol chemical production was increased:**

Thanks for the reviewer's suggestion. In this manuscript, aerosol chemistry (AERC) process refers to microphysical nucleation, condensation, and coagulation, as well as the mass transfer between the gas phase and condensed phase.

The increased 24-h $PM_{2.5}$ change (23:00LST minus 00:00LST) due to AERC (+29.6 μg m$^{-3}$) over Beijing-Tianjin-Hebei (BTH) during the haze formation stage (December 16-22, Stage_1) was mainly attributed to the production of secondary inorganic aerosols, e.g., sulfate ($SO_4^{2-}$, +2.7 μg m$^{-3}$), nitrate ($NO_3^-$, +16.8 μg m$^{-3}$), and ammonium ($NH_4^+$, +9.8 μg m$^{-3}$).

In order to explain why AERC was larger in Stage_1 than that in Stage_2 (December23-27, aerosol dispersion stage), two metrics (sulfur oxidation ratio (SOR), $SOR = nSO_4^{2-}/(nSO_4^{2-} + nSO_2)$, and nitric oxidation ratio (NOR), $NOR = nNO_3^-/(nNO_3^- + nNO_2)$, n refers to the molar concentration) can be used to estimate the degree of secondary formation of $SO_4^{2-}$ and $NO_3^-$ (Sun et al., 2006; Zhao et al., 2013). Figure R1 shows the time series of calculated SOR and NOR averaged over the BTH region from 16 to 29 December 2015. When SOR and NOR are less than 0.1, $SO_4^{2-}$ and $NO_3^-$ mainly come from the primary source emissions, otherwise, high

oxidation rates of SOR and NOR can result in large fractions of $SO_4^{2-}$ and $NO_3^-$ in $PM_{2.5}$ (Fu et al., 2008). The mean SOR and NOR in Stage_1 (0.25 and 0.23, respectively) are larger than that in Stage_2 (0.23 and 0.19, respectively), indicating more aerosol particles can be produced in Stage_1.

[Figure]

**Figure R1.** Time series of SOR (sulfur oxidation ratio, shown in red) and NOR (nitric oxidation ratio, shown in blue) averaged over the BTH (Beijing-Tianjin-Hebei) region from 16 to 29 December 2015.

**Response to why vertical mixing removal was decreased:**

In WRF-Chem, dry deposition (DRYD) is intermingled with vertical turbulent diffusion (DIFF), and the sum of DRYD and DIFF can reflect the role of vertical mixing (VMIX) in relocating the airborne pollutants vertically (Tao et al., 2015).

[Figure]

**Figure R2.** Differences in vertical profiles of daily equivalent potential temperature (EPT, shown in blue), EPT in the daytime (09:00LST-17:00LST, shown in red), and EPT at night (18:00LST-08:00LST, shown in black) between Stage_1 (16-22 December) and Stage_2 (23-27 December) averaged over the BTH region.

Under general conditions, VMIX can take place due to any inhomogeneous heating of the atmosphere (Li et al., 2017), which means VMIX is influenced by atmospheric stability, and the atmosphere stability can be exactly characterized by the profile of equivalent potential temperature (EPT) (Bolton, 1980; Zhao et al., 2013; Yang et al., 2016). If EPT rises with height, then the atmosphere is stable.

Figure R2 shows the differences in vertical profiles of EPTs between Stage_1 and Stage_2 averaged over the BTH region. The calculated daily EPT, ETP in the daytime (09:00LST-17:00LST) and EPT at night (18:00LST-08:00LST) in Stage_1 are all larger than that in Stage_2, and the differences are increased with the height (below ~400 m for the daily difference, below ~500 m in the daytime, and below ~300 m at night). All these indicate that the atmosphere is more stable during the pollution accumulation period, and the stabilized atmospheric layer will weaken VMIX to make more air pollutants accumulate in the near-surface layer in BTH.

2. *In Section 4.1, the authors used changes in synoptic conditions and atmospheric circulation to explain the aerosol variation, which seems not consistent with the IPR results.*

**Response:**

Thanks for the reviewer's suggestion. The developed IPR (integrated process rate) analysis scheme adopted in this manuscript is used to quantify the contributions of each physical/chemical process to the variation in $PM_{2.5}$ concentrations. We mainly focus on the pollution accumulation (dispersion) period to reveal important factors that cause the increase (decrease) in concentrations of aerosol particles. So firstly, we should reproduce the evolution of the daily $PM_{2.5}$ concentrations during 16-29 December 2015, aiming to divide the severe haze event into aerosol accumulation stage and aerosol dispersion stage according to the average $PM_{2.5}$ concentration in BTH, which can be significantly influenced by atmospheric circulation and synoptic conditions.

3. *The specific humidity did not have a visible change during Dec.20-22, so the aerosol chemical production was not due to the humidity. Probably it was due to increases in aerosol precursors from regional transport.*

**Response:**

We totally agree with the reviewer's opinion. Figure R3 shows the diurnal variation of $PM_{2.5}$ concentrations averaged over the BHT region during 20-22 December 2015. The $PM_{2.5}$ concentration is increased by 50.4 μg m$^{-3}$ (from 237.0 μg m$^{-3}$ at 00:00LST to 287.4 μg m$^{-3}$ at 23:00LST), and the contribution of AERC to the 24-h $PM_{2.5}$ increase is +43.8 μg m$^{-3}$, which contains the contributions from aerosol chemical productions of $SO_4^{2-}$ (+2.6 μg m$^{-3}$), $NO_3^-$ (+27.0 μg m$^{-3}$), and $NH_4^+$ (+14.1 μg m$^{-3}$).

[Figure]

**Figure R3.** Diurnal variations of $PM_{2.5}$ concentrations averaged over the BTH (Beijing-Tianjin-Hebei) region during 20-22 December 2015 (shown by purple dot line). The colored bars represent different components.

In this manuscript, AERC refers to microphysical nucleation, condensation, and coagulation, as well as the mass transfer between the gas phase and condensed phase. So the increased concentrations of $SO_4^{2-}$, $NO_3^-$ and $NH_4^+$ due to AERC may mainly come from the gas-particle partition. As shown in Fig. R1, higher oxidation rates of SOR and NOR are calculated during December 20-22, indicating more $SO_4^{2-}$ and $NO_3^-$ can be produced from the gas-phase and/or liquid-phase oxidation of $SO_2$ (sulfur dioxide) and $NO_2$ (nitrogen dioxide).

4. *Although the model reproduced well the PM$_{2.5}$ variation during the haze event, it strongly underestimated concentrations of most aerosols, but overestimated nitrate concentration (Figure 5). However, during stage_1, it looks that most PM$_{2.5}$ increase is due to nitrate increase (Figure 8a). The authors need to discuss the potential influences of model biases on the results.*

**Response:**

Thanks for the reviewer's suggestion. The biases in BC (black carbon) may due to the uncertainty in primary emissions (Li et al., 2017). Secondary organic aerosols are not considered in the MOSAIC (Model for Simulating Aerosol Interactions and Chemistry) aerosol module (Qiu et al., 2017), which makes the simulated OC (organic carbon) concentrations lower. Missing oxidation mechanisms (including gas-phase and aqueous-phase oxidation, as well as heterogeneous chemistry) of $SO_2$ may result in the under-predicted sulfate concentration, which also allows for excess nitrate in the presence of ammonia (Gao et al., 2014; Chen et al., 2016). Meanwhile, there may be an issue with $NO_x$ partitioning and/or missing $NO_x$ sink in current air quality models (Chen et al., 2019), making WRF-Chem overpredict the nitrate concentration. It should be noted that similar biases can also be found in other WRF-Chem studies (Zhang et al., 2015; Qiu et al., 2017).

Although underestimation/overestimation of aerosol compositions are simulated by the WRF-Chem model at the sites in Beijing (39.97 °N, 116.37 °E) and in Shijiazhuang (38.03 °N, 114.53 °E), the predicted total PM$_{2.5}$ surface concentrations in each city capture the observations pretty well (Fig. 4), especially the evolution characteristics during the whole simulation period. What's more, the main purpose of this manuscript is to investigate the evolution mechanisms of the total PM$_{2.5}$ concentrations during a haze event over BTH, including the contributions of each detailed physical or chemical process to the variations in the total PM$_{2.5}$ concentration.

According to the reviewer's suggestion, a discussion has been added in the revised manuscript as follows "The uncertainty of the MIX anthropogenic emission inventory, the lack of secondary organic aerosols, and the missing mechanisms of some heterogeneous reactions may result in large uncertainties in the final simulation results, especially the predicted aerosol chemical compositions, such as $SO_4^{2-}$, $NO_3^-$ and $NH_4^+$. The biases in simulated concentrations of $SO_4^{2-}$, $NO_3^-$ and $NH_4^+$ may have impacts on the contributions of AERC and CLDC processes to the air pollution variation. Uncertainties should be quantitatively analyzed in future studies". **(Page 20, Line 4-8)**

**Specific Comments:**
1. *Page 1 Line 23: Delete 'happened'.*
**Response:**

According to the reviewer's suggestion, we have deleted it in the revised manuscript.

*2. Page 1 Line 24 and following parts: Please change 'outside transport' to 'regional transport'.*
**Response:**
    Thanks for your suggestion. We have changed the expression in the whole revised manuscript.

*3. Page 3 Line 12: Delete 'even'.*
**Response:**
    According to the reviewer's suggestion, we have deleted it in the revised manuscript.

*4. Page 4 Line 5: Change 'contribution' to 'contributor'.*
**Response:**
    According to the reviewer's suggestion, we have changed the expression in the revised manuscript. **(Page 4, Line 4)**

*5. Page 4 Line 14: Recent studies found that black carbon-East Asian winter monsoon interactions and dust-wind interactions can also intensify winter haze in eastern China (e.g., Yang et al., 2017; Lou et al., 2019). The authors may would like to cite these studies.*
**Response:**
    Thanks for the reviewer's suggestion. Related references have been added in the revised manuscript.
    "These impacts are coupled with atmospheric dynamics to produce a chain of interactions with a large range of meteorological variables that influence both weather and climate (Ramanathan et al., 2001; Huang et al., 2006; Li et al., 2017c; Yang et al., 2017) , which will further induce feedbacks on aerosol production, accumulation, and even severe haze pollutions (Petaja et al., 2016; Li et al., 2017d; Zhao et al., 2017; Gao et al., 2018; Lou et al., 2019)". **(Page 4, Line 10-13)**
    "The light-absorbing aerosols can also amplify haze in NCP by weakening East Asian winter monsoon wind speeds through ocean and cloud feedbacks (Lou et al., 2019)". **(Page 4, Line 17-19)**

*6. Page 8 Line 19: What does emission source mean and how it affects aerosol variation with 24-hr (Figure 8). Does the model include diurnal variation of emission? Why contributions of EMIS are different between stage_1 and stage_2?*
**Response to what does emission source mean and how it affects aerosol variation with 24-hr (Figure 8):**
    Emission source (EMIS) means the primary emissions from anthropogenic emission, biogenic emission, biomass-burning emission, dust emission, sea-salt emission and so on. The emitted aerosols from primary emission sources can increase the near-surface $PM_{2.5}$ concentrations significantly.

**Response to does the model include diurnal variation of emission:**
    According to Chen et al. (2019), the diurnal and weekly variations of anthropogenic emission factors for each sector (power, industry, residential and transportation) are adopted in this

manuscript (Fig. R4).

**Response to why contributions of EMIS are different between stage_1 and stage_2:**

Thanks for your suggestion. The daily EMIS in Stage_1 and Stage_2 should be the same, and we have recalculated the 24-h values (261.2 μg m$^{-3}$) during the adjusted simulation period (Stage_1: December 16-22; Stage_2: December 23-27).

[Figure]

**Figure R4.** (a) Diurnal and (b) weekly variations of anthropogenic emission factors for power, industry, residential and transportation sectors.

*7. Page 9 Line 19: Change 'closing' to 'turning off'.*
**Response:**

According to the reviewer's suggestion, we have changed the expression in the revised manuscript. **(Page 9, Line 18)**

*8. Page 11 Line 4: How these matrices calculated?*
**Response:**

According to the reviewer's suggestion. The detailed calculation methods about the statistics are described in Table 3 in the revised manuscript. **(Page 30)**

*9. Page 14 Line 14-22: Please rephrase this paragraph by illustrating absolute change first then percentage change (relative to what?). And what are the rest of contribution, from natural emission or emission outside the domain?*
**Response:**

Thanks for your suggestion. In Fig.7, the unit of absolute contribution is "μg m$^{-3}$", and the absolute contribution of local emission (regional transport) to the PM$_{2.5}$ concentration in BTH can be expressed as the difference between OnlyBTH_Anth and NoAnth (the difference between NoBTH_Anth and NoAnth). For percentage contribution or relative contribution, the unit is "%", and it can be calculated by dividing the absolute contribution by the simulated PM$_{2.5}$ concentration from CTL. Meanwhile, according to the comments from Reviewer#2, the expressions of "absolute PM$_{2.5}$ concentrations" have been changed to "PM$_{2.5}$ concentrations" in the revised manuscript.

The rest of the contribution (represented by "Others" in Fig.7 ) to the PM$_{2.5}$ concentrations in

BTH may include the impacts from biogenic emission, biomass-burning emission, natural emission, aerosols outside the simulation domain, and the non-linear chemical formation for secondary aerosol.

According to the reviewer's suggestion, we have revised the paragraph as follows "As shown in Fig. 7(a), the $PM_{2.5}$ concentration in BTH during Stage_1 was mainly contributed by the combined effects of local emission and regional transport. The contributions of local emission and regional transport to the $PM_{2.5}$ concentration were comparable (49% and 32%, respectively), especially during the heavy pollution period (December 20-22, 43% vs. 37%). During Stage_2, the contributions of regional transport decreased from 30% to 16%. The relative high $PM_{2.5}$ concentration (107.9 $\mu g$ $m^{-3}$) was principally caused by the local emission. On average, the contributions of local emission and regional transport to the $PM_{2.5}$ concentration in Stage_2 were 51% and 24%, respectively. The impact of regional transport could be qualitatively expressed by specific humidity, which was treated as an indicator for the origin of air masses (Jia et al., 2008). Air masses from the south were usually warmer and wetter than those from the north, so the specific humidity averaged over the BTH was higher in Stage_1 (1.7 g/kg) than that in Stage_2 (1.4 g/kg) (Fig. 7(b)). The evolution of $PM_{2.5}$ nicely followed the trend of specific humidity with a high correlation coefficient of 0.86.". **(Page 14-15, Line 19-28)**

*10. Page 15 Line 7: Please clarify that the dominant sources of surface-layer $PM_{2.5}$ 'variation'.*
**Response:**

Thanks for the reviewer's suggestion. Figure S3 shows the weighted contributions of each physical/chemical process to hourly $PM_{2.5}$ changes in Stage_1 and Stage_2. The weighted contributions can be calculated using the equation $\%PC_i = \frac{PC_i}{\sum_{i=1}^{n}|PC_i|}$ (Goncalves et al., 2009), where $PC_i$ is the absolute contribution ($\mu g$ $m^{-3}$) (i.e., the change in $PM_{2.5}$ concentration induced by process *i*), and $\%PC_i$ is the weighted contribution (%) of process *i*. Note that the sum of $\%PC_i$ for all processes may not be 100%, but the sum of $abs(\%PC_i)$ is exactly 100%.

From Fig. S3, we can find that EMIS and AERC are the major contributors to make the $PM_{2.5}$ concentration increase. But the processes of TRAN (advection), DRYD (dry deposition) and DIFF (turbulent diffusion) make the $PM_{2.5}$ concentration decrease. The impacts of other processes (e.g., wet scavenging (WETP), cloud chemistry (CLDC)) on the hourly $PM_{2.5}$ variation can be negligible ($-5\% < \%PC_i < 5\%$).

In the early morning (00:00LST-05:00LST) in Stage_1 and Stage_2, TRAN is the major negative contributor to make the $PM_{2.5}$ concentration decrease. But at noon (10:00LST-13:00LST), the major negative contributor is DIFF. During 07:00LST-08:00LST and 16:00LST-21:00LST, EMIS is the major positive contributor to make the $PM_{2.5}$ concentration increase. Generally, TRAN, EMIS and DIFF are the dominant processes which affect the surface-layer $PM_{2.5}$ variation.

*11. Figure 7: Change 'special' to 'specific' in caption and figure.*
**Response:**

Thanks for your suggestion. We have revised it in the whole revised manuscript.

*12. Figure 11: Change '18h' to '20h'.*

**Response:**

According to the reviewer's suggestion, we have changed the label in the revised figure. **(Page 41)**

[revised manuscript text omitted]
$^{-3}$) | 59 | 210.0 | 194.3 | -15.7 | 79.2 | -7.5 | 110.0 | 2.8 | 44.3 | 0.7 | 0.8 |

| Variables | nstd[b] | OBS[b] | SIM[b] | NMB[3b] | MFB[4b] | MFE[5b] | IOA[6b] | R[7b] |
|---|---|---|---|---|---|---|---|---|
| T$_2$ (k)[a] | 12 | 270.7 | 271.6 | 10.3 | 10.3 | 10.7 | 0.94 | 0.90 |
| RH$_2$ (%)[a] | 12 | 63.8 | 56.1 | -12.1 | -121.8 | 22.2 | 0.82 | 0.73 |
| WS$_{10}$ (m s$^{-1}$)[a] | 12 | 2.5 | 3.2 | 28.3 | 32.4 | 587.5 | 0.798 | 0.707 |
| WD$_{10}$ (°)[a] | 12 | 190.8 | 192.2 | 10.8 | -21.6 | 554.8 | 0.657 | 0.43 |
| PM$_{2.5}$ (μg m$^{-3}$) | 59 | 173.6 | 168.2 | -3.1 | 132.7 | 47.3 | 0.869 | 0.768 |

[a]T$_2$: temperature at 2 m (k); RH$_2$: relative humidity at 2 m (%); WS$_{10}$: wind speed at 10 m (m s$^{-1}$); WD$_{10}$: wind direction at 10 m (°).

[b]nstd: the number of observation sites; [1,2]$\overline{OBS}$ and $\overline{SIM}$ represent the average observations and simulations, respectively. OBS: the average observations; $\overline{OBS} = \frac{1}{nstd} \times \sum_{i=1}^{nstd} OBS_i$, SIM: the average simulations $\overline{SIM} = \frac{1}{nstd} \times \sum_{i=1}^{nstd} SIM_i$.

MB: mean bias; GE: gross error; [3]NMB is the NMB: normalized mean bias (%). $NMB = \frac{1}{nstd} \times \sum_{i=1}^{nstd} \frac{SIM_i - OBS_i}{OBS_i} \times 100\%$.

RMSE: root mean square error; [4]MFB is the MFB: mean fractional bias (%). $MFB = \frac{2}{nstd} \times \sum_{i=1}^{nstd} \frac{SIM_i - OBS_i}{SIM_i + OBS_i} \times 100\%$.

[5]MFE is the MFE: mean fractional error (%). $MFE = \frac{2}{nstd} \times \sum_{i=1}^{nstd} \frac{|SIM_i - OBS_i|}{SIM_i + OBS_i} \times 100\%$.

[6]IOA is the IOA: index of agreement. $IOA = 1 - \frac{\sum_{i=1}^{nstd}(SIM_i - OBS_i)^2}{\sum_{i=1}^{nstd}(|OBS_i - \overline{OBS}| + |SIM_i - \overline{SIM}|)^2}$.

[7]R is the R: correlation coefficient. $R = \frac{\sum_i^{nstd}|(OBS_i - \overline{OBS}) \times (SIM_i - \overline{SIM})|}{\sqrt{\sum_i^{nstd}(OBS_i - \overline{OBS})^2 + \sum_i^{nstd}(SIM_i - \overline{SIM})^2}}$.

[revised manuscript text omitted]
{OBS} = \frac{1}{nstd} \times \sum_{i=1}^{nstd} OBS_i$, $\overline{SIM} = \frac{1}{nstd} \times \sum_{i=1}^{nstd} SIM_i$.

[3]NMB is the normalized mean bias, $NMB = \frac{1}{nstd} \times \sum_{i=1}^{nstd} \frac{SIM_i - OBS_i}{OBS_i} \times 100\%$.

[4]MFB is the mean fractional bias, $MFB = \frac{2}{nstd} \times \sum_{i=1}^{nstd} \frac{SIM_i - OBS_i}{SIM_i + OBS_i} \times 100\%$.

[5]MFE is the mean fractional error, $MFE = \frac{2}{nstd} \times \sum_{i=1}^{nstd} \frac{|SIM_i - OBS_i|}{SIM_i + OBS_i} \times 100\%$.

[6]IOA is the index of agreement, $IOA = 1 - \frac{\sum_{i=1}^{nstd}(SIM_i - OBS_i)^2}{\sum_{i=1}^{nstd}(|OBS_i - \overline{OBS}| + |SIM_i - \overline{SIM}|)^2}$.

[7]R is the correlation coefficient, $R = \frac{\sum_i^{nstd}|(OBS_i - \overline{OBS}) \times (SIM_i - \overline{SIM})|}{\sqrt{\sum_i^{nstd}(OBS_i - \overline{OBS})^2 + \sum_i^{nstd}(SIM_i - \overline{SIM})^2}}$.

[revised manuscript text omitted]

---

## Author Comment (AC2) · 4 Jul 2019

**Response to Comments of Reviewer #2**

**(comments in *italics*)**

**Manuscript number:** acp-2019-245

**Title:** Assessing the formation and evolution mechanisms of severe haze pollution in Beijing-Tianjin-Hebei region by using process analysis

Review of: "Assessing the formation and evolution mechanisms of severe haze pollution in Beijing–Tianjin–Hebei region by using process analysis" by Chen et al.

*The authors quantitatively examined the cause of a severe haze event over Beijing–Tianjin–Hebei (BTH) through decomposing contributions from emissions, physical and chemical processes, using the WRF–Chem model equipped with an improved integrated process rate (IPR) analysis scheme. The IPR scheme also reveals the dominant role of aerosol radiative effects in haze formation is physical rather than chemical. Such an IPR scheme merits application in future relevant studies. Overall, the manuscript is well structured and should be of great interest to ACP readers. I recommend it is publishable after my minor comments/suggestions are addressed.*

**Response:**

Thanks to the reviewer for the valuable comments and suggestions which are very helpful for us to improve our manuscript. We have revised the manuscript carefully, as described in our point-to-point responses to the comments.

**General Comments:**

1. ***Local vs. regional contributions:*** *From the sensitivity simulations, local emissions and regional transport account for 80% of total $PM_{2.5}$. What does the contribution from "others" (Fig.7) mean? Is it because the non-linear chemical formation for secondary aerosol or contribution from aerosol precursors outside of BTH?*

**Response:**

According to the experiments listed in Table 2, the contributions of local anthropogenic emission and regional transport to the $PM_{2.5}$ concentrations in BTH (Beijing-Tianjin-Hebei) can be identified by comparing the simulation results between OnlyBTH_Anth and NoAnth (i.e., OnlyBTH_Anth minus NoAnth), and between NoBTH_Anth and NoAnth (i.e., NoBTH_Anth minus NoAnth), respectively.

In addition to the primary source emission (e.g., anthropogenic emission) and regional transport, secondary aerosol formation and their hygroscopic growth are also considered to be a large contributor to severe haze episodes (Huang et al., 2014b; Han et al., 2015; Chen et al., 2019).

As Li et al., (2018) pointed out that a "brute-force" method (e.g., sensitivity analysis used to measure the model outputs response to emission change) is a traditional way to identify source contributions from non-reactive species in a linear process, but it cannot straightforwardly apply to secondary species due to the non-linearity in responses. All these indicate that the actual impact of one factor in a nonlinear process in the presence of others can be separated into (1) pure impact

from the factor, and (2) interactional impacts from other factors.

In this manuscript, we divide the simulated PM$_{2.5}$ concentration into three parts: contributions from local anthropogenic emission, regional transport and others. Besides the impacts of local anthropogenic emissions and regional transport, the rest of the contributions to the PM$_{2.5}$ concentration in BTH may include the impacts from biogenic emission, biomass-burning emission, dust emission, sea salts, aerosols outside the simulation domain, and the non-linear chemical formation for secondary aerosol.

2. *Advection contribution: A negative value for advection is diagnosed by the IPR scheme. From my understanding, the advection means horizontal transport, which should not be always a negative contribution, and instead it may contribute a lot to PM$_{2.5}$ if taking the value of regional contribution as an equivalent.*

**Response:**

We totally agree with the reviewer's opinion. In this manuscript, TRAN includes horizontal and vertical advection, which is highly related to wind and aerosol concentration gradients from upwind regions to downwind areas (Gao et al., 2018). When the calculated TRAN in a model grid during a simulation output interval is negative, indicating the process of advection will decrease the aerosol concentrations, and vice versa.

[Figure]

**Figure R1.** (a-n) Spatial distribution of simulated daily transport fluxes of PM$_{2.5}$ during 16-29 December 2015. The average PM$_{2.5}$ flux in Stage_1 (December 16-22) and Stage_2 (December 23-27) are also shown in (o) and (p).

Figure R1 shows the spatial distributions of simulated transport fluxes of PM$_{2.5}$ during 16-29

December 2015. The average flux in Stage_1 (December 16-22) and Stage_2 (December 23-27) are also shown. During Stage_1 (Fig. R1(o)), the southerly wind over the southern parts of BTH is low, resulting in weak $PM_{2.5}$ fluxes from polluted upstream regions (e.g., Henan and Shandong) to downstream regions (e.g., BTH). However, pollutants in the northern parts of BTH are transported eastwardly to the regions of Huanghai and Bohai Sea. Generally, a negative value of TRAN is diagnosed by the integrated process rate (IPR) scheme in Stage_1. Similar influence of TRAN on $PM_{2.5}$ concentrations in BTH can also be found in Stage_2. This is because prevailing northerly winds in BTH bring aerosols to downstream regions (e.g., Henan and Shandong).

3. *Aerosol radiative effects: It is considerate to include the aerosol indirect effect, though this process contributes marginally in less-cloud wintertime. But the authors failed to show/discuss how aerosol indirect effect is expressed from IPR result. For example, in Fig. 11 the CLDC (0.5 ug m$^{-3}$) and WETP (0.2 ug m$^{-3}$) can be taken as the result of aerosol indirect effect. It needs to clarify.*

**Response:**

Following the comments from Reviewer#3, simulation results from 16 to 18 December 2015 are also considered in this manuscript to better analyze the formation mechanism of the haze event. According to the time series of simulated daily $PM_{2.5}$ concentrations in BTH (Fig. 6(l)), aerosol accumulation stage (Stage_1) is now considered during December 16-22. All the values in Stage_1 have been re-calculated in the revised manuscript, and now, the CLDC and WETP in Fig. 11 are 0.5 ug m$^{-3}$ and 0.2 ug m$^{-3}$, respectively.

According to the reviewer's comments, another sensitivity experiment (referred to as NoAIE case) is designed. Same as CTL, but a prescribed vertically uniform cloud droplet number ($0.93 \times 10^8$ particles per kg of air) is used in NoAIE, which is calculated from the CTL case during the whole simulation period, following Gao et al., (2015) and Zheng et al., (2015). Comparing the simulation results from CTL and NoAIE, the impacts of aerosol indirect effects (AIEs) on haze episode can be analyzed and quantified.

**Table R1.** Contributions of each process to the change in 24-h $PM_{2.5}$ increase caused by aerosol indirect effects during Stage_1. The 24-h increase in $PM_{2.5}$ concentration (23:00LST minus 00:00LST) in CTL and NoAIE are 43.9 ug m$^{-3}$ and 44.2 ug m$^{-3}$, respectively.

| Process | TRAN | EMIS | VMIX | SGCV | GASC | CLDC | AERC | WETP | $PM_{2.5}$ |
|---------|------|------|------|------|------|------|------|------|-----------|
| CTL-NoAIE (ug m$^{-3}$) | 0.3 | 0.0 | 1.2 | 0.0 | 0.0 | 0.4 | -2.3 | 0.1 | -0.3 |

From Table R1, we can find that when AIE is considered, the 24-h increase of $PM_{2.5}$ concentration during Stage_1 is decreased by 0.3 ug m$^{-3}$, from 44.2 ug m$^{-3}$ in NoAIE to 43.9 ug m$^{-3}$ in CTL. The reduction induced by AIE can be mainly attributed to the contribution of aerosol chemistry (AERC, -2.3 ug m$^{-3}$). AERC refers to microphysical nucleation, condensation, and coagulation, as well as the mass transfer between the gas phase and condensed phase. As Zheng et al., (2015) pointed out that when AIE is included, the predicted cloud droplet number is based on the simulated aerosol number different than what is prescribed in the model default setting. This coding strategy allows interstitial air-borne aerosols to become cloud-borne aerosols after activation.

Therefore, the simulated PM$_{2.5}$ concentrations are decreased.

The change induced by AIE in contributions of vertical mixing (VMIX) process during Stage_1 is +1.2 ug m$^{-3}$, this is because PBLH (planetary boundary layer height) is reduced, and the decreased PBLH can result in the accumulation of aerosol particles in the near-surface layer.

Due to low cloud cover (or small cloud water content) and little precipitation in northern China in winter as shown in Zhao et al. (2015) and Zheng et al., (2015), the contributions of CLDC (cloud chemistry, +0.4 ug m$^{-3}$) and WETP (wet scavenging, +0.1 ug m$^{-3}$) induced by AIE to the 24-h PM$_{2.5}$ change are relative small.

**Specific comments:**

The presentation of this work would be greater if some editorial aspects are improved.

1. *Use of %: P1L27. Here and elsewhere in the text, please round off 250% instead of 250.0%. The decimal doesn't make sense in terms of model bias.*

**Response:**

According to the reviewer's suggestion, the decimal fraction has been rounded off in the whole revised manuscript.

2. *Please check the appropriate usage of hyphen (-) (e.g., near-surface) and en-dash (–) (e.g., December 20–22).*

**Response:**

Thanks for your suggestion. The hyphen has been used in the whole revised manuscript.

3. *Fig.1: The purple dot (?) for PBLH is hardly to see. Reduce the size of green triangle or increase its transparency.*

**Response:**

According to the reviewer's suggestion, Figure 1 has been re-plotted in the revised manuscript. **(Page 31)**

4. *Fig.4: I suggest the use of NMB and correlation coefficient are good enough for model evaluation. Reader gets lost in so many numbers.*

**Response:**

According to the reviewer's suggestion, several statistics (e.g., mean bias (MB), gross error (GE), and root mean square error (RMSE)) have been removed in the revised manuscript.

5. *Fig.6: I can't see any different for (k) and (i), and there is also no discussion in the text. Remove one of them.*

**Response:**

Thanks for your suggestion. Only the time series of simulated daily PM$_{2.5}$ concentrations averaged over the BTH region are shown in Fig. 6(l) in the revised manuscript. **(Page 36)**

6.  *Fig.8: Move the middle panel (b1 and b2) towards right. Is there any difference in Y-axis of (b1) and (b2). I suppose they are the same.*
**Response:**
Thanks for your suggestion. The Y coordinates in Figs. 8(b1) and (b2) are the same, and we have re-plotted the figure in the revised manuscript. **(Page 38)**

7.  *P1L24: There is any special meaning for "absolute" PM2.5. If not, please "absolute" when it is unnecessary.*
**Response:**
According to the reviewer's suggestion, the expressions of "absolute $PM_{2.5}$ concentrations" have been changed to "$PM_{2.5}$ concentrations" in the revised manuscript.

8.  *P2L6 & P5L8: Remove "and so on".*
**Response:**
According to the reviewer's suggestion, we have deleted it in the revised manuscript.

9.  *P3L7: I don't think severe haze frequently occurs in wintertime over PRD region, and neither of your two references support this.*
**Response:**
Thanks for the reviewer's suggestion. We have deleted it in the revised manuscript.

10. *P3L10-12: Health threatened by PM2.5 is the most important thing people care about.*
**Response:**
We totally agree with the reviewer's opinion. Observations show that annual $PM_{2.5}$ concentrations in China are more than 5 times higher than the World Health Organization (WHO) guideline value in some metropolitans (Wang et al., 2014). Sustained exposure to high $PM_{2.5}$ concentrations greatly threatens public health (Hu et al., 2014; Wang et al., 2015; Burnett et al., 2018), including lung cancer (Dockery et al., 1993), cardiopulmonary disease (Pope and Dockery, 2006), bronchitis (Gao et al., 2015) and so on. Source sector contributions of anthropogenic emissions to complex air pollution and their health impacts will be discussed in our upcoming study.

11. *P5L1: Remove "(SPM)". You don't use it in the following text.*
**Response:**
According to the reviewer's suggestion, we have deleted it in the revised manuscript.

12. *P5L3: Haze is not actually caused by "the interactions between …". It's a synergy effect by these factors.*

**Response:**

According to the reviewer's suggestion, we have revised the sentence as follows "All these studies discussed above revealed that the formation of haze episode was caused by the synergy impacts of local emissions, regional transport, meteorological conditions, and chemical production". **(Page 5, Line 3-4)**

13. *P5L12-13: Is there any reference saying "substantial efforts since 2009"? Zheng et al. (2018, ACP, Trends in China's anthropogenic emissions since 2010 as the consequence of clean air actions) shows emissions dropped substantially only after 2013.*

**Response:**

Thanks for the reviewer's comments. Since the 9th FYP (Five-Year Plan, 1996-2000), the policies about the emission controls on gases (e.g., $CO_2$ and $SO_2$) and energy saving were initiated (Cao et al., 2009). In the 11th FYP (2006-2010), an obligatory target about the emission reductions for each local government was further outlined (Anger et al., 2016), and the policies were maintained and extended in the 12th FYP (Jin et al., 2016). Although the implementations of total control policies were successful and the targets were achieved, the air quality improvement was insignificant (Schreifelds et al., 2012).

A severe haze event happened in January 2013 over many provinces in China. This haze with its unprecedentedly high index of $PM_{2.5}$ concentrations and extremely low visibility was of worldwide concern. Quickly responding to the $PM_{2.5}$ crisis, the Chinese government issued the well-known action plan "China National Action Plan on Air Pollution Prevention and Control" in September 2013, which means the "war" against air pollution was declared. Since then, emissions of multi-pollutants are reduced (Zheng et al., 2018) and industrial structures are optimized (Bao and Yao, 2016).

According to the reviewer's suggestion, we have revised the sentence as follows "Since 2013, substantial efforts have been taken to improve air quality in China, including emission reduction and energy transition". **(Page 5, Line 12-13)**

14. *P7L2: Which year of anthropogenic emission from MIX?*

**Response:**

The anthropogenic emissions in China are taken from the monthly 2010 Multi-resolution Emission Inventory (MEIC, http://www.meicmodel.org/). According to Chen et al. (2019), the diurnal and weekly variations of anthropogenic emission factors for each sector (power, industry, residential and transportation) are also adopted in this manuscript (Fig. R2).

[Figure]

**Figure R2.** (a) Diurnal and (b) weekly variations of anthropogenic emission factors for power, industry, residential and transportation sectors.

15. *P8L20: Sub-grid convection (SGCV) is found to be zero in the simulation. It can be possibly due to no precipitation in this case. But why gas-phase chemistry is also zero, or what it specifically means? Nitrate formation is dominantly through gas-phase oxidation of NO2+OH.*

**Response:**

Thanks for the reviewer's suggestion. SGCV (sub-grid convection) refers to the scavenging within the sub-grid wet convective updrafts. The simulated SGCV in the CTL case is zero. This is because the daily accumulated cumulus precipitation averaged in BTH during 16-29 December 2015 is zero.

GASC (gas-phase chemistry) is simulated by CBMZ (Carbon-Bond Mechanism version Z), and this process mainly focuses on the reactions between gases, including the gas phase photooxidation reactions. Detailed parameterizations can be found in Zaveri and Peters (1999). Take gas phase $HNO_3$ as an example, the principle formation path for $HNO_3$ (g) in the daytime is $NO_2 + OH \rightarrow HNO_3$. At nighttime, $N_2O_5$ gas can react with vapor phase of $H_2O$ (gas-phase reaction, the heterogeneous reaction of $N_2O_5$ ($N_2O_5 + H_2O$ (l) $\rightarrow 2HNO_3$(l)) is treated by the aerosol scheme) to form $HNO_3$ (g).

In this manuscript, the IPR analysis is applied to diagnose the contributions of physical/chemical processes to the variations in aerosol concentrations (e.g., $SO_4^{2-}$, $NO_3^-$, $NH_4^+$ and $PM_{2.5}$). This is why the calculated contributions of GASC to $PM_{2.5}$ variations in Fig. 8 are zero.

16. *P9L14 & L16: You haven't defined what "two stages" are. Maybe don't need to mention this here.*

**Response:**

Thanks for your suggestion. We have deleted it, and the section of numerical experiments (Section 2.3) has be revised as follows:

"Table 2 summarizes the experimental designs. To investigate the contributions of regional transport and local emission to the $PM_{2.5}$ concentrations in BTH, four simulations with different anthropogenic emission categories are conducted: (1) CTL: The control simulation with all anthropogenic emissions considered; (2) NoAnth: No anthropogenic emission is considered in the whole domain; (3) NoBTH_Anth: Same as CTL, but anthropogenic emissions in BTH are excluded; (4) OnlyBTH_Anth: Contrary to the NoBTH_Anth case, anthropogenic emissions are only

considered in BTH. All the physical and chemical schemes used in these cases are identical. The contributions of regional transport and local emission to the PM$_{2.5}$ concentration in BTH can be identified by comparing the simulation results of NoBTH_Anth and NoAnth (i.e., NoBTH_Anth minus NoAnth) and OnlyBTH_Anth and NoAnth (i.e., OnlyBTH_Anth minus NoAnth), respectively". **(Page 9, Line 9-16)**

"To quantify the aerosol radiative effects (ARE) on haze pollution, another sensitivity experiment (referred to as NoARE case) is designed by turning off the feedbacks between aerosols and meteorological variables, including eliminating the aerosol direct effect (ADE) and aerosol indirect effect (AIE) in the model. The ADE is turned off by removing the mass of aerosol species from the calculation of aerosol optical properties as did in Qiu et al. (2017). The AIE is turned off by using a prescribed vertically uniform cloud droplet number, which is calculated from the CTL case during the whole simulation period, following Gao et al. (2015) and Zhang et al., (2015a). The differences between CTL and NoARE (i.e., CTL minus NoARE) represent the impacts of aerosol radiative forcing". **(Page 9, Line 17-23)**

"The IPR analysis method is applied to all the designed experiments. Comparing the contributions of each detailed process between pollution accumulation stage and dissipation stage in CTL can quantitatively explain the reason for the variation of the PM$_{2.5}$ concentrations in BTH. Meanwhile, the prominent physical or chemical process responsible for the aerosol radiative impacts on the haze episode can also be investigated by analyzing the IPR analysis method used in CTL and NoARE cases". **(Page 9-10, Line 24-28)**

"All the five simulations are conducted for the period from 13 to 29 December 2015, and the initial three days are discarded as the model spin-up to minimize the impacts of initial conditions. Simulation results from the CTL case during 16 to 29 December 2015 are used to evaluate the model performance". **(Page 10, Line 5-7)**

*17. P10L12-19: Please delete unnecessary "marked in" in this paragraph.*
**Response:**

We have revised the paragraph (Section 2.4) according to the reviewer's suggestion.

"Simulated meteorological parameters in CTL case, including 2 m temperature (T$_2$), 2 m relative humidity (RH$_2$), 10 m wind speed (WS$_{10}$) and 10 m wind direction (WD$_{10}$), are compared with hourly observations at twelve stations, which are collected from NOAA's National Climatic Data Center (https://gis.ncdc.noaa.gov/maps/ncei/cdo/hourly). Due to limited observations of PBL height in BTH, the retrieved PBLH in 3-hour intervals obtained from the GDAS (Global Data Assimilation System) (https://ready.arl.noaa.gov/READYamet.php) in Beijing (39.93 °N, 116.28 °E) is also used to evaluate the model performance. More detailed information about the GDAS meteorological dataset (1 °×1 °) can be found in Rolph et al. (2013), Kong et al. (2015) and https://www.ready.noaa.gov/gdas1.php. Hourly shortwave downward radiation flux (SWDOWN) at the Xianghe station (39.75 °N, 116.96 °E) is taken from WRMC-BSRN (World Radiation Monitoring Center-Baseline Surface Radiation Network, http://bsrn.awi.de) for the energy budget evaluation. The hourly observed surface-layer PM$_{2.5}$ concentrations at the 59 stations are obtained from the CNEMC (China National Environmental Monitoring Center, http://www.cnemc.cn/). The daily measurements of mass concentrations of SO$_4^{2-}$, NO$_3^-$, NH$_4^+$, BC and OC are collected at the sites of (39.97 °N, 116.37 °E) in Beijing and (38.03 °N, 114.53 °E) in Shijiazhuang (Huang et al., 2017;

Liu et al., 2018). Detailed locations of these observations are shown in Fig. 1(b)". **(Page 10, Line 9-21)**

*18. P11L4-6 & P12L10-11: The NMB (IOA) and R are the only metrics you used and they are informative enough. I suggest remove other metrics if possible. Or define them in Table 3.*

**Response:**

According to the reviewer's suggestion, several statistics (e.g., mean bias (MB), gross error (GE), and root mean square error (RMSE)) have been removed in the revised manuscript. Meanwhile, the detailed calculation methods about the statistics are described in Table 3 in the revised manuscript.

*19. P11L10: Change "options" to "parameterizations"*

**Response:**

According to the reviewer's suggestion, we have revised it in the manuscript. **(Page 11, Line 12)**

*20. P13L24: Add "air quality" before "threshold value"*

**Response:**

According to the reviewer's suggestion, the sentence has been revised in the manuscript. **(Page 14, Line 3)**

*21. P14L8: I suggest change "internal" to "dominant" or "leading"*

**Response:**

Thanks for your comments, and we have changed it in the revised manuscript.

"Previous studies have reported that anthropogenic emission was the dominant cause of haze events in China (Jiang et al., 2013; Sun et al., 2014; Gu and Liao, 2016; Yang et al., 2016b)". **(Page 14, Line 12-13)**

*22. P16L2-4: Please delete "Suspended ... during winter haze periods". You have done this in the Introduction section.*

**Response:**

According to the reviewer's suggestion, we have revised the paragraph as follows "Previous studies have demonstrated that the aerosol radiative forcing could increase the near-surface $PM_{2.5}$ concentrations by about 12%-29% (Gao et al., 2015; Gao et al., 2016; Qiu et al., 2017; Zhou et al., 2018). However, the detailed influence mechanisms (i.e., the prominent physical or chemical process responsible for the aerosol radiative impacts on $PM_{2.5}$ concentrations) are still unclear. In this section, we examine the effects of aerosol radiative forcing on meteorological parameters and $PM_{2.5}$ levels during the haze episode, with a special focus on the detailed influence mechanism by using the IPR analysis". **(Page 16, Line 5-10)**

*23. P16L15-16: Is this process important in your case? If not, saying this here reads misleading.*

**Response:**

Thanks for your suggestion. The sentence has been deleted in the revised manuscript.

*24. P16L10: I suggest to move Fig. S4 in the main text. It is interesting to show the important role of absorbing aerosol on regional circulation change. This result is consistent with the simulation by Qiu et al. (2017).*

**Response:**

Thanks for your suggestion. The main purpose of this manuscript is to investigate the formation and evolution mechanisms of a haze event in BTH during 16-29 December 2015, including examining the contributions of local emission and regional transport to the $PM_{2.5}$ concentration in BTH, and the contributions of each detailed physical or chemical process to the variations in the $PM_{2.5}$ concentration. The influence mechanisms of aerosol radiative forcing (including aerosol direct and indirect effects) are also examined by using the process analysis. Figure S4 is only used to explain why aerosol radiative effects had a negative impact on the near-surface $PM_{2.5}$ concentrations in BTH during December 23-24. So we decide to leave the figure in the supplement.

From Fig. S4, we can find that an anomalous northeasterly induced by absorbing aerosols was simulated, leading to a decrease in the near-surface $PM_{2.5}$ concentrations, which is different from previous studies that reported light-absorbing aerosols could worsen air quality (Li et al., 2016; Huang et al., 2018; Gao et al., 2018).

We totally agree with the reviewer's opinion about the important roles of absorbing aerosols on regional circulation changes. More experiments will be designed to examine the changes in atmospheric thermal and atmospheric dynamic caused by absorbing aerosol radiative forcing and their effects on haze episodes in our future studies.

*25. Conclusions and discussions. The "conclusions" part can be shortened and concise, which should make room for more insightful discussion. I came up some ideas. (1) how the IPR scheme can be further improved? (2) the authors could discuss the possible application of IPR scheme in future haze study (both winter and summer), because the work provides a quantitative analysis of how aerosol radiative effects change $PM_{2.5}$ through physical and chemical pathways.*

**Response:**

According to the reviewer's suggestion, the words in the conclusion part have been cut back by ~20% in the revised manuscript, and more discussions have been added in the last section, including some limitations in the current study and several possible applications of the IPR analysis in future studies. Here are the revised paragraphs:

"There are some limitations in this work. The uncertainty of the MIX anthropogenic emission inventory, the lack of secondary organic aerosols, and the missing mechanisms of some heterogeneous reactions may result in large uncertainties in the final simulation results, especially the predicted aerosol chemical compositions, such as $SO_4^{2-}$, $NO_3^-$ and $NH_4^+$. The biases in simulated concentrations of $SO_4^{2-}$, $NO_3^-$ and $NH_4^+$ may have impacts on the contributions of AERC and CLDC processes to the air pollution variation. Uncertainties should be quantitatively

analyzed in future studies. Besides, conclusions draw from a case study in BTH cannot represent a full view of the underlying mechanisms of haze formation and elimination. Better understanding will be attained by conducting multiple-case simulations in future. What's more, an anomalous northeasterly induced by absorbing aerosols was observed, leading to a decrease in the near-surface $PM_{2.5}$ concentrations during December 23-24 2015 in BTH, which was different from previous studies that reported light-absorbing aerosols could worsen air quality (Li et al., 2016; Huang et al., 2018; Gao et al., 2018). More experiments should be designed in future to examine the changes in atmospheric thermal and atmospheric dynamic caused by absorbing aerosol radiative forcing and their impacts on haze episodes". **(Page 20, Line 4-15)**

"As Zheng et al. (2018) pointed out that the $PM_{2.5}$ concentration in China has been decreasing in recent years, but the decreased fine particulate matter could stimulate ozone production (Li et al., 2019a; Zhu et al., 2019). Multi-pollutant mixture may be a hot topic in the future, and the IPR analysis can be a useful method to provide a quantitative analysis about the formation mechanism of the complex air pollutions, including figuring out the major physical/chemical process behind these events. Meanwhile, significant differences between model predictions (e.g., $O_3$ and $PM_{2.5}$) are found among current multi-scale air quality models (Chen et al., 2019b; Li et al., 2019b), even though the same inputs are used. These different performances can be associated with the differences in model formulations, including parameterizations and numerical methods (Carmichael et al., 2008). In order to acquire a quantitative attribution of the cause of differences between simulation results, process analysis method should be developed and implemented in these models, and the IPR analysis will be easier to draw conclusions about the fundamental problems that cause the differences between model predictions". **(Page 20, Line 16-25)**

[revised manuscript text omitted]
{OBS} = \frac{1}{nstd}\times\sum_{i=1}^{nstd} OBS_i$, $\overline{SIM} = \frac{1}{nstd}\times\sum_{i=1}^{nstd} SIM_i$.

NMB is the normalized mean bias, $NMB = \frac{1}{nstd}\times\sum_{i=1}^{nstd}\frac{SIM_i-OBS_i}{OBS_i}\times100\%$.

MFB is the mean fractional bias, $MFB = \frac{2}{nstd}\times\sum_{i=1}^{nstd}\frac{SIM_i-OBS_i}{SIM_i+OBS_i}\times100\%$.

MFE is the mean fractional error, $MFE = \frac{2}{nstd}\times\sum_{i=1}^{nstd}\frac{|SIM_i-OBS_i|}{SIM_i+OBS_i}\times100\%$.

IOA is the index of agreement, $IOA = 1 - \frac{\sum_{i=1}^{nstd}(SIM_i-OBS_i)^2}{\sum_{i=1}^{nstd}(|OBS_i-\overline{OBS}|+|SIM_i-\overline{SIM}|)^2}$.

R is the correlation coefficient, $R = \frac{\sum_i^{nstd}|(OBS_i-\overline{OBS})\times(SIM_i-\overline{SIM})|}{\sqrt{\sum_i^{nstd}(OBS_i-\overline{OBS})^2+\sum_i^{nstd}(SIM_i-\overline{SIM})^2}}$.

Where $OBS_i$ and $SIM_i$ mean observations and model predictions, respectively. $i$ refers to a given station, and nstd is the total number of stations.

[Figure]

**Figure 1.** (a) Map of the two nested model domains. (b) Locations of the observations used for model evaluation.

[Figure]

[Figure]

**Figure 2. Time series of observed (shown in black dots) and simulated (shown in red dots) hourly 2 m temperature (T2, k), 2 m relative humidity (RH2, %), 10 m wind speed (WS10, m s-1), and 10 m wind direction (WD10, ˚) averaged over the 12 stations during 2016 -29 December 2015.**

[Figure]

[revised manuscript text omitted]
{OBS} = \frac{1}{nstd} \times \sum_{i=1}^{nstd} OBS_i$, $\overline{SIM} = \frac{1}{nstd} \times \sum_{i=1}^{nstd} SIM_i$.

[3]NMB is the normalized mean bias, $NMB = \frac{1}{nstd} \times \sum_{i=1}^{nstd} \frac{SIM_i - OBS_i}{OBS_i} \times 100\%$.

[4]MFB is the mean fractional bias, $MFB = \frac{2}{nstd} \times \sum_{i=1}^{nstd} \frac{SIM_i - OBS_i}{SIM_i + OBS_i} \times 100\%$.

[5]MFE is the mean fractional error, $MFE = \frac{2}{nstd} \times \sum_{i=1}^{nstd} \frac{|SIM_i - OBS_i|}{SIM_i + OBS_i} \times 100\%$.

[6]IOA is the index of agreement, $IOA = 1 - \frac{\sum_{i=1}^{nstd}(SIM_i - OBS_i)^2}{\sum_{i=1}^{nstd}(|OBS_i - \overline{OBS}| + |SIM_i - \overline{SIM}|)^2}$.

[7]R is the correlation coefficient, $R = \frac{\sum_{i}^{nstd}|(OBS_i - \overline{OBS}) \times (SIM_i - \overline{SIM})|}{\sqrt{\sum_{i}^{nstd}(OBS_i - \overline{OBS})^2 + \sum_{i}^{nstd}(SIM_i - \overline{SIM})^2}}$.

[revised manuscript text omitted]

---

## Author Comment (AC3) · 4 Jul 2019

**Response to Comments of Reviewer #3**

**(comments in *italics*)**

**Manuscript number:** acp-2019-245

**Title:** Assessing the formation and evolution mechanisms of severe haze pollution in Beijing-Tianjin-Hebei region by using process analysis

**General Comments:**

*This study investigated a haze event over BTH region in December 2015 with the process analysis method. The study examined the mechanisms underlying the event formation and evolution. They found the event was mainly controlled by the change of vertical mixing. In the end, the study also found that the vertical mixing and transport were two main processes that were responsible for the aerosol radiative feedback. The manuscript is well-written. However, the main point of processing analysis alone is not novel at all. Many previous studies have used this method in multiple air quality models, including WRF-Chem. The study only selected one event as the analysis case. Although the study found that the vertical mixing was the main contributor to the formation and evolution of the event, I didn't find anything new brought to the community. Besides these general comments, I also have some specific comments.*

**Response:**

Thanks to the reviewer for the valuable comments and suggestions which are very helpful for us to improve our manuscript.

Process analysis techniques (i.e., integrated process rate (IPR) analysis) (Gipson, 1999) have already been fully implemented in CMAQ model (Community Multi-scale Air Quality model), and this air quality model with the IPR analysis has been widely used to analyze the formation and evolution mechanisms of ozone episodes (Goncalves et al., 2009; Khiem et al., 2010; Xing et al., 2017; Tang et al., 2017) and aerosol pollutions (Liu et al., 2011; Yang and Shiang-Yuh, 2013; Fan et al., 2014). However, the IPR analysis has not yet been officially adopted in the WRF-Chem model (Weather Research and Forecasting-Chemistry model) (Tao et al., 2015). Although several WRF-Chem model studies have used the method to investigate the impacts of physical/chemical processes on variations in $O_3$ concentrations (Jiang et al., 2012; Gao et al., 2018), few studies conducted the IPR analysis with WRF-Chem for aerosols.

Meanwhile, China has been suffering from serious haze pollutions, especially in the North China Plain in winter (Han et al., 2014; Gao et al., 2015; Sun et al., 2016). Extensive studies have been carried out to investigate the formation mechanisms underlying haze episodes (Liu et al., 2017; Wang et al., 2017; Kang et al., 2019; Wu et al., 2019), and try to explore possible solutions to improve air quality (Chen et al., 2019; Wang et al., 2019). But conclusions from previous studies can only reflect the combined effects of all physical and chemical processes. Detailed information of the impacts of individual process on haze events is usually unavailable.

Therefore, we develop an improved IPR analysis scheme in the WRF-Chem model to isolate nine physical/chemical processes impacting variations in aerosol concentrations and track their contributions quantitatively. The nine different processes are advection (TRAN), emission source (EMIS), dry deposition (DYRD), turbulent diffusion (DIFF), sub-grid convection (SGCV), gasphase chemistry (GASC), cloud chemistry (CLDC), aerosol chemistry (AERC), and wet scavenging (WETP). We then use the IPR analysis to investigate the formation and evolution mechanisms of a severe haze event over Beijing-Tianjin-Hebei (BTH) during 16 to 29 December 2015, including analyzing the influence mechanisms of aerosol radiative forcing. Some results can be summarized as follows: (1) the $PM_{2.5}$ increase over BTH during the haze formation stage (December 16 to 22) may be mainly attributed to strong production by aerosol chemistry process and weak removal by advection and vertical mixing processes; (2) the restrained vertical mixing can be the primary reason for the enhancement in near-surface $PM_{2.5}$ increase when aerosol radiative forcing is considered.

In our future studies, multiple haze events during 2013 to 2018 will be further analyzed by using the IPR analysis to figure out the major physical/chemical process behind these pollutions.

**Specific Comments:**

1. *The study selected the event of Dec. 20-29. Based on the result, it seems that the $PM_{2.5}$ concentration reached 200 μg/m³ on average on Dec. 20. To better show the formation of event, the simulation and analysis should start from the earlier date to demonstrate the concentration rising from a lower level such as 50 μg/m³ as shown in the end of this event.*

**Response:**

According to the reviewer's suggestion, simulation period has been extended from 13 to 29 December 2015, and the simulation results during December 16-29 are analyzed. Figure R1 shows the time series of simulated daily $PM_{2.5}$ concentrations averaged over the BTH region during December 16-29. The $PM_{2.5}$ concentration over BTH in December 16 is 24.2 μg m$^{-3}$.

Due to the updated simulation period, all the results have been re-calculated in the revised manuscript. Here shows the revised abstract "Fine-particle pollution associated with haze threatens human health, especially in the North China Plain, where extremely high $PM_{2.5}$ concentrations were frequently observed during winter. In this study, the WRF-Chem model coupled with an improved integrated process analysis scheme was used to investigate the formation and evolution mechanisms of a haze event over Beijing-Tianjin-Hebei (BTH) in December 2015, including examining the contributions of local emission and regional transport to the $PM_{2.5}$ concentration in BTH, and the contributions of each detailed physical or chemical process to the variations in the $PM_{2.5}$ concentration. The influence mechanisms of aerosol radiative forcing (including aerosol direct and indirect effects) were also examined by using the process analysis. During the aerosol accumulation stage (December 16-22, Stage_1), the near-surface $PM_{2.5}$ concentration in BTH was increased from 24.2 μg m$^{-3}$ to 289.8 μg m$^{-3}$, with the contributions of regional transport increased from 12% to 40%, while the contributions of local emission were decreased from 59% to 38%. During the aerosol dispersion stage (December 23-27, Stage_2), the average concentration of $PM_{2.5}$ was 107.9 μg m$^{-3}$, which was contributed by local emission of 51% and regional transport of 24%. The 24-h change (23:00LST minus 00:00LST) in the near-surface $PM_{2.5}$ concentration was +43.9 μg m$^{-3}$ during Stage_1 and -41.5 μg m$^{-3}$ during Stage_2. Contributions of aerosol chemistry, advection and vertical mixing to the 24-h change were +29.6 (+17.9) μg m$^{-3}$, -71.8 (-103.6) μg m$^{-3}$ and -177.3 (-221.6) μg m$^{-3}$ during Stage_1 (Stage_2), respectively. Small differences in contributions of other processes were found between Stage_1 and Stage_2. Therefore, the $PM_{2.5}$ increase over BTH during haze formation stage was mainly attributed to the strong production by aerosol chemistry process and weak removal by advection and vertical mixing processes. When aerosol radiative feedback was

considered, the 24-h PM$_{2.5}$ increase was enhanced by 4.8 μg m$^{-3}$ during Stage_1, which could be mainly attributed to the contributions of vertical mixing process (+22.5 μg m$^{-3}$), advection process (-19.6 μg m$^{-3}$) and aerosol chemistry process (+1.2 μg m$^{-3}$). The restrained vertical mixing was the primary reason for the enhancement in near-surface PM$_{2.5}$ increase when aerosol radiative forcing was considered". **(Page 1-2)**

[Figure]

**Figure R1.** Time series of simulated daily PM$_{2.5}$ concentrations averaged over the Beijing-Tianjin-Hebei region from 16 to 29 December 2015.

2.  *Line 19-21 of page 6, if the FDDA is turned on for the control and noARE experiments, I do expect some aerosol meteorological feedbacks can be diminished by the FDDA. Free runs without FDDA is preferred for studying aerosol feedback.*

**Response:**

Thanks for the reviewer's suggestion, and we totally agree with your opinion. Following Feng et al. (2016), Mar et al. (2016) and Werner et al. (2016), four-dimensional data assimilation (FDDA) is only applied to the first domain in this manuscript, and no analysis nudging is included for the inner second domain.

Here is the revised sentence "Four-dimensional data assimilation (FDDA) with the nudging coefficient of 3.0×10$^{-4}$ for wind (in and above PBL), temperature (above PBL) and water vapor mixing ratio (above PBL) is adopted to improve the accuracy of simulation results (no analysis nudging is included for the inner domain) (Lo et al., 2008; Otte, 2008; Wang et al., 2016b; Werner et al., 2016)". **(Page 6, Line 21-23)**

3.  *Line 11-12 of page 7, the MOSAIC aerosol mechanism in WRF-Chem has not been coupled with the Shao dust emission scheme, at least in the publicly released version. If it was coupled in this study or any previous studies, please briefly introduce it and cite the related references.*

**Response:**

Thanks for the reviewer's suggestion. The Shao 2004 dust emission scheme (referred to as Shao_2004) is proposed by Shao (2004) and implemented in WRF-Chem by Kang et al. (2011) and Wu and Lin (2013). Previous studies have reported that Shao_2004 scheme shows a good performance in dust emission amounts over source areas, including the spatial distribution of dust particles over the downwind regions over East Asia (Wu and Lin, 2013; Kang et al., 2014; Su and Fung, 2015).

Three important parameters are used to calculate the dust emission amounts: (1) the threshold friction velocity, (2) the horizontal sand flux, and (3) the vertical dust flux. The threshold friction

velocity is defined as the minimum friction velocity to initiate soil particle movement. It can be parameterized by the cohesive force which is proportional to particle size (Shao and Lu, 2000), and it is strongly affected by soil moisture, salt concentrations in the soil, and roughness elements on the surface. The horizontal sand flux indicates the intensity of dust saltation, defined as a vertical integral of the streamwise saltating particle flux density when the friction velocity exceeds the threshold friction velocity (White, 1979). The vertical dust flux is defined as the emitted dust mass concentration per unit area per unit time, and it is calculated using the equation proposed by Shao (2004). More detailed descriptions of the dust parameterization can be found in Shao (2004) and Kang et al. (2011).

Although in the publicly released version of the WRF-Chem model, the MOSAIC (Model for Simulating Aerosol Interactions and Chemistry) aerosol mechanism has not been coupled with the Shao_2004 scheme, we have tried to make several modifications in the script of module_mosaic_addemiss.F (i.e., add a new subroutine to process the Shao dust emission scheme), aiming to equip the sectional aerosol mechanism with the new Shao_2004 scheme. Following Zhao et al. (2010), similar parameters are used to partition the total dust masses into eight bins in the MOSAIC module.

Analyzing the simulation results from Chen et al. (2018), the WRF-Chem model (the MOSAIC aerosol mechanism has been coupled with the Shao dust emission scheme) can successfully reproduce the spatiotemporal evolution of a dust storm happened during 14-17 April 2015.

4. *Line 18-24 of page 9, this part is confusing. In WRF-Chem, the aerosol-cloud interaction is linked with wet deposition and cloud aqueous chemistry. If aerosol-cloud interaction is turned off in this study, then the CTL and noARE experiments should use different chemistry mechanisms, i.e., noARE likely used the one without wet deposition and cloud chemistry. Please provide more details about this. If this is the case, the difference between the control and noARE should include not only the aerosol feedback but also the difference in chemical processes. Based on the results, it seems that there were little cloud and precipitation during the period. The major aerosol feedback is from aerosol-radiation interaction, therefore, it makes more sense that in noARE experiment only the aerosol radiative feedback is turned off with aerosol-cloud interaction not touched. Furthermore, we generally do not call aerosol-cloud interaction as aerosol radiative effects.*

**Response:**

We totally agree with the reviewer's comments that aerosol-cloud interactions include the cloud chemistry and wet scavenging.

In the WRF-Chem model, aerosol indirect effects (AIEs) (including the first and the second indirect effects) are implemented by linking simulated droplet number to radiation schemes (e.g., Rapid Radiative Transfer Model for Global Circulation Models (RRTMG) shortwave radiation scheme) and microphysics schemes (e.g., Purdue Lin Scheme) (Skamarock et al., 2008). The Lin scheme can predict the cloud droplet number (Chen and Sun, 2002), and the auto-conversion is dependent on it, following Liu et al. (2005). Aerosol particles acting as cloud condensation nuclei are coupled with the cloud physics portion of the model. This coupling allows for fully interactive feedbacks: aerosols affect cloud droplet number and cloud radiative properties, and clouds alter aerosol size and composition via aqueous processes and wet scavenging (Gustafson et al., 2007).

In order to access the impacts of aerosol-cloud interactions, the most common approach is to design a hypothetical scenario (i.e., NoAIE) with a prescribed distribution of cloud droplet number (Zhao et al., 2017), and the aerosol indirect effect (AIE) can be quantified by comparing the simulations results between the baseline scenario and NoAIE.

Following the suggestion provided by the WRF-Chem user's guide (https://ruc.noaa.gov/wrf/wrf-chem/Users_guide.pdf), AIE is turned off in this manuscript by using a prescribed vertically uniform cloud droplet number, which is calculated from the control case during the whole simulation period. Similar processing method can also be found in many other studies (Forkel et al., 2015; Kong et al., 2015; Zhang et al., 2015; Zhao et al., 2017).

5. *Line 11 of page 10, please provide the full name of NOAA READY GDAS. In addition, please provide more information about the PBL data from this dataset. Is it retrieval or direct observation? If it is retrieval, what is the method used for the retrieval?*

**Response:**

Thanks for your suggestion. The full names of NOAA, READY and GDAS are National Oceanic and Atmospheric Administration (NOAA), Real-time Environmental Applications and Display sYstem (READY), and Global Data Assimilation System (GDAS), respectively.

The meteorological data of planetary boundary layer height (PBLH) is extracted from the National Oceanic and Atmospheric Administration's (NOAA) Global Data Assimilation System (GDAS) data. As shown by Huang et al. (2012), PBL heights of GDAS agree well with the vertical lidar observations in Shanghai.

The National Weather Service's National Centers for Environmental Prediction (NCEP) runs a series of computer analyses and forecasts operationally. One of the operational systems is GDAS (Nemuc et al., 2012). GDAS can be assimilated by surface observations, balloon data, wind profiler data, aircraft reports, buoy observations, radar observations, and satellite observations (Rolph et al., 2017). GDAS is run four times a day at 00:00, 06:00, 12:00, and 18:00UTC. Model outputs include the analyses and forecast fields at three hours after each analysis (Wang et al., 2014). NCEP post-processing of the GDAS converts the data from spectral coefficient form to 1 degree latitude-longitude grids and from sigma levels to 23 pressure layers. More detailed information can be found at https://www.ready.noaa.gov/gdas1.php.

Generally, PBLH is calculated every 3-hour each day by the NOAA's READY Archived Meteorology online calculating program (http://ready.arl.noaa.gov/READYamet.php). This program can plot a time-series of calculated boundary layer depth using the chosen meteorological data. The calculations use the same equations as the NOAA HYSPLIT (Hybrid Single-Particle Lagrangian Integrated Trajectory) transport and dispersion model.

In HYSPLIT, there are two options to estimate the boundary layer stability. The preferred method is to use the fluxes of heat and momentum provided by the meteorological model, if available. Otherwise the temperature and wind gradients of each grid-point sounding are used to estimate stability. More information about the calculation method is described by Draxler and Rolph (2003).

According to the reviewer's suggestion, we have revised the sentence as follows "Due to limited observations of PBL height in BTH, the retrieved PBLH in 3-hour intervals obtained from the GDAS (Global Data Assimilation System) (https://ready.arl.noaa.gov/READYamet.php) in

Beijing (39.93 °N, 116.28 °E) is also used to evaluate the model performance. More detailed information about the GDAS meteorological dataset (1 °×1 °) can be found in Rolph et al. (2013), Kong et al. (2015) and https://www.ready.noaa.gov/gdas1.php". **(Page 10, Line 11-15)**

6. *In the figures of hourly timeseries such as Fig. 3 or 4, please specify whether it is local time or UTC time? In Fig. 3, is there a low limit from PBL retrieval? It seems the values are limited to 50 m. The same is applied to the simulations. Any specific reason?*

**Response:**

 Thanks for the reviewer's suggestion. The UTC time has been converted to China Standard Time by adding 8 h to Beijing Time in China. Related descriptions have been added in the revised figures (Fig. 3 and 4). **(Page 33-34)**

 At night, the PBL height collapses into a shallow stable boundary layer, retrieved and simulated PBLHs are close to ~50 m and ~30 m, respectively. The low limit of the PBLH is chosen to correspond to the minimum height resolution (Draxler and Hess, 2004).

7. *Fig. 5 shows the aerosol components at the station of Shijiazhuang? Why not show the total PM2.5 surface concentration at this station as a reference? In addition, since the simulation seems capturing the hourly PM2.5 variation well, why not show the hourly component comparison instead of period average only? It would be interesting and provide useful information.*

**Response:**

 Thanks for the reviewer's suggestion. The daily measurements of mass concentrations of $SO_4^{2-}$, $NO_3^-$, $NH_4^+$, BC and OC are collected at the observation site (38.03 °N, 114.53 °E) in the city of Shijiazhuang, and these observations are provided by the Campaign on Atmospheric Aerosol Research network of China (CARE-China). More information about these chemical observations can be found in Huang et al. (2017) and Liu et al. (2018).

 We totally agree with the reviewer's opinion, but we can only get the daily concentrations of these aerosol chemical compositions ($SO_4^{2-}$, $NO_3^-$, $NH_4^+$, BC and OC) at the two sites ((39.97 °N, 116.37 °E) and (38.03 °N, 114.53 °E)). What's more, similar evaluation method can also be found in other studies (Gao et al., 2016; Qiu et al., 2017).

8. *What does the black line represent in Fig. 10?*

**Response:**

 The black line in Fig. 10 means the zero contour line, which can be used to clearly distinguish the positive and negative differences. According to the reviewer's suggestion, we have revised the caption of Fig. 10 as follows "Figure 10. Time series of differences in (a) temperature (k), (b) equivalent potential temperature (k), (c) vertical wind speed (cm s$^{-1}$), (d) relative humidity (%), and (e) PM$_{2.5}$ concentration ($\mu$g m$^{-3}$) between CTL and NoARE cases (CTL minus NoARE) averaged over the Beijing-Tianjin-Hebei region. The purple and green lines denote the simulated PBLH in CTL and NoARE cases, respectively. The black line represents the zero contour line". **(Page 40)**

9. *In Fig. 11, the process analysis showed the averaged 24-h change of PM$_{2.5}$ during the period. What does this mean? Why is the averaged 24-h change important? I think that the change through each stage of the event would be more interesting. Please clarify.*

**Response to the question of "In Fig. 11, the process analysis showed the averaged 24-h change of PM2.5 during the period. What does this mean?"**

Simulation results from 16 to 29 December 2015 are analyzed in this manuscript. According to the daily PM$_{2.5}$ concentrations averaged over the BTH region (Fig. 6(l)), the life cycle of the haze event typically consists of the two stages: (1) aerosol accumulation stage (December 16-22, Stage_1), and (2) aerosol dispersion stage (December 23-27, Stage_2).

Take the first stage as an example, simulated hourly PM$_{2.5}$ concentrations in BTH during December 16-22 are averaged into one day to show the mean diurnal variation of the near-surface PM$_{2.5}$ concentrations, as listed in Figs. 11(a) and (b).

Analyzing Figs. 11(a) and (b), we can find that the PM$_{2.5}$ concentration is increased by 39.1 μg m$^{-3}$ in the NoARE case, from 127.4 μg m$^{-3}$ at 00:00LST to 166.5 μg m$^{-3}$ at 23:00LST. However, the 24-h change is larger (+43.9 μg m$^{-3}$) in CTL case, from 136.5 μg m$^{-3}$ at 00:00LST to 180.4 μg m$^{-3}$ at 23:00LST. All these indicate that when aerosol radiative effects (ARE) are considered, the diurnal evolution of the simulated near-surface PM$_{2.5}$ concentrations is enhanced (+39.1 μg m$^{-3}$ vs. +43.9 μg m$^{-3}$), which means more suspended aerosol particles will deteriorate the air quality in the next day in BTH.

In order to explain the enhancement of 4.8 μg m$^{-3}$ (12%) induced by ARE, the IPR analysis is used to the track the contributions of each physical/chemical process, as shown in Fig. 11(c), and the restrained vertical mixing can be the primary reason for this enhancement.

**Response to the question of "Why is the averaged 24-h change important? I think that the change through each stage of the event would be more interesting. Please clarify."**

Thanks for the reviewer's suggestion. If the change of the PM$_{2.5}$ concentrations through the entire aerosol accumulation stage is used to investigate the evolution mechanism of the haze event, only the values at the begging time (00:00LST in December 16) and the finish time (23:00LST in December 22) are analyzed. Many useful information will be discarded, especially the daily evolution characteristics of the simulated PM$_{2.5}$ concentrations in BTH.

In this manuscript, hourly PM$_{2.5}$ concentrations during each stage are averaged into one day to show the mean diurnal variation of the near-surface PM$_{2.5}$ concentration. Similar analytical method can also be found in many other studies (Fan et al., 2015; Tao et al., 2015; Xing et al., 2017).

10. *In Fig. 12b, using height (m) instead of model levels as the y-axis makes more sense.*
**Response:**

According to the reviewer's suggestion, we have re-plotted the figure in the revised manuscript.
**(Page 42)**

**Thank you very much for your comments and suggestions.**

**Marked-up Manuscript:**

[revised manuscript text omitted]

MB: mean bias; GE: gross error; [3]NMB is the NMB: normalized mean bias (%), $NMB = \frac{1}{nstd} \times \sum_{i=1}^{nstd} \frac{SIM_i - OBS_i}{OBS_i} \times 100\%$.

RMSE: root mean square error; [4]MFB is the MFB: mean fractional bias (%), $MFB = \frac{2}{nstd} \times \sum_{i=1}^{nstd} \frac{SIM_i - OBS_i}{SIM_i + OBS_i} \times 100\%$.

[5]MFE is the MFE: mean fractional error (%), $MFE = \frac{2}{nstd} \times \sum_{i=1}^{nstd} \frac{|SIM_i - OBS_i|}{SIM_i + OBS_i} \times 100\%$.

10  [6]IOA is the IOA: index of agreement, $IOA = 1 - \frac{\sum_{i=1}^{nstd}(SIM_i - OBS_i)^2}{\sum_{i=1}^{nstd}(|OBS_i - \overline{OBS}| + |SIM_i - \overline{SIM}|)^2}$.

[7]R is the R: correlation coefficient, $R = \frac{\sum_{i}^{nstd}|(OBS_i - \overline{OBS}) \times (SIM_i - \overline{SIM})|}{\sqrt{\sum_{i}^{nstd}(OBS_i - \overline{OBS})^2 + \sum_{i}^{nstd}(SIM_i - \overline{SIM})^2}}$.

[revised manuscript text omitted]
{OBS} = \frac{1}{nstd} \times \sum_{i=1}^{nstd} OBS_i$, $\overline{SIM} = \frac{1}{nstd} \times \sum_{i=1}^{nstd} SIM_i$.

[3]NMB is the normalized mean bias, $NMB = \frac{1}{nstd} \times \sum_{i=1}^{nstd} \frac{SIM_i - OBS_i}{OBS_i} \times 100\%$.

[4]MFB is the mean fractional bias, $MFB = \frac{2}{nstd} \times \sum_{i=1}^{nstd} \frac{SIM_i - OBS_i}{SIM_i + OBS_i} \times 100\%$.

[5]MFE is the mean fractional error, $MFE = \frac{2}{nstd} \times \sum_{i=1}^{nstd} \frac{|SIM_i - OBS_i|}{SIM_i + OBS_i} \times 100\%$.

[6]IOA is the index of agreement, $IOA = 1 - \frac{\sum_{i=1}^{nstd}(SIM_i - OBS_i)^2}{\sum_{i=1}^{nstd}(|OBS_i - \overline{OBS}| + |SIM_i - \overline{SIM}|)^2}$.

[7]R is the correlation coefficient, $R = \frac{\sum_{i}^{nstd}|(OBS_i - \overline{OBS}) \times (SIM_i - \overline{SIM})|}{\sqrt{\sum_{i}^{nstd}(OBS_i - \overline{OBS})^2 + \sum_{i}^{nstd}(SIM_i - \overline{SIM})^2}}$.

[revised manuscript text omitted]